# Human mobility data and machine learning reveal geographic differences in alcohol sales and alcohol outlet visits across U.S. states during COVID-19

**Yingjie Hu**[1]*, **Brian M. Quigley**[2], **Dane Taylor**[3]

**1** Department of Geography, University at Buffalo, The State University of New York, Buffalo, NY, United States of America, **2** Department of Medicine, University at Buffalo, The State University of New York, Buffalo, NY, United States of America, **3** Department of Mathematics, University at Buffalo, The State University of New York, Buffalo, NY, United States of America

* yhu42@buffalo.edu

## Abstract

As many U.S. states implemented stay-at-home orders beginning in March 2020, anecdotes reported a surge in alcohol sales, raising concerns about increased alcohol use and associated ills. The surveillance report from the National Institute on Alcohol Abuse and Alcoholism provides monthly U.S. alcohol sales data from a subset of states, allowing an investigation of this potential increase in alcohol use. Meanwhile, anonymized human mobility data released by companies such as SafeGraph enables an examination of the visiting behavior of people to various alcohol outlets such as bars and liquor stores. This study examines changes to alcohol sales and alcohol outlet visits during COVID-19 and their geographic differences across states. We find major increases in the sales of spirits and wine since March 2020, while the sales of beer decreased. We also find moderate increases in people's visits to liquor stores, while their visits to bars and pubs substantially decreased. Noticing a significant correlation between alcohol sales and outlet visits, we use machine learning models to examine their relationship and find evidence in some states for likely panic buying of spirits and wine. Large geographic differences exist across states, with both major increases and decreases in alcohol sales and alcohol outlet visits.

## Introduction

When the pandemic of COVID-19 quickly spread in the United States in March 2020, stay-at-home orders were implemented in many states to reduce the transmission of the coronavirus. As millions of Americans were confined at home, a combination of factors, including the fear of contracting the disease, social isolation, job loss, and the uncertain future, created a facilitating environment for increased problematic alcohol use [1]. Anecdotes from various sources suggested a surge in alcohol sales as the pandemic began [2–4]. Some early investigations suggested an increase in alcohol use during COVID-19 shutdowns [5], and alcohol was classified

**Data Availability Statement:** The alcohol sales data used in this study can be downloaded from https://pubs.niaaa.nih.gov/publications/surveillance-covid-19/COVSALES.htm. The human

mobility data are from SafegGraph, and one can request access at: https://www.safegraph.com/.

**Funding:** The authors received no specific funding for this work.

**Competing interests:** The authors have declared that no competing interests exist.

as an essential product and remained available during the shutdown of many states [6]. Meanwhile, other reports suggested no major changes or even a decline in alcohol sales, citing reasons such as misconception [7, 8]. A preliminary study in the UK showed that there were about 20% of individuals who increased their normal alcohol consumption during the COVID-19 lockdown, but there were also a similar number of individuals who decreased their normal alcohol consumption during the same lockdown period [9].

These mixed reports raise the question as to whether there was indeed a surge in alcohol sales in the U.S. as the stay-at-home orders began. Answering this question is important because widespread increases in alcohol use, if left unattended, can increase the susceptibility of the public to COVID-19 [1] and lead to many other negative societal consequences. Increased alcohol use may be especially likely for those experiencing anxiety and depression during the pandemic [10–13]. Alcohol use is associated with a number of adult health-related consequences [14–17] and negative effects on infant development when the infant is exposed to alcohol in the womb [18, 19]. Heavy alcohol use in young adults is also linked to numerous social problems, including driving under the influence [20], bar violence [21], domestic violence [22], and other forms of public disorder [23]. Understanding the changes in alcohol sales associated with the pandemic can help policymakers address social problems related to alcohol use and better prepare for future public health crises.

The National Institute on Alcohol Abuse and Alcoholism (NIAAA) provides monthly alcohol sales data for a number of individual U.S. states. According to NIAAA, these data were collected by various state sources primarily for taxation purposes. While not covering all U.S. states, this dataset provides sales information about spirits, wine, and beer, respectively, thereby enabling an analysis of the sales changes for different types of alcohol. This dataset does not distinguish between on- and off-premise alcohol outlets, and it provides only total monthly sales in a state. Although having its limitations, this dataset is highly valuable as it allows researchers to quantitatively measure the changes of alcohol sales during the COVID-19 pandemic. NIAAA has published a Surveillance Report examining gross changes in alcohol sales from pre to post initiation of the pandemic [24]; however, a detailed examination of this data in combination with other data sources, such as human mobility data, has not yet been conducted to better understand the factors that account for pandemic-related changes in alcohol sales.

Anonymized human mobility data collected from smart mobile devices (e.g., smartphones) have strong potential to complement the data released by government agencies such as NIAAA. Since the pandemic began, companies such as Descartes Lab, SafeGraph, Google, Facebook, Foursquare, PlaceIQ, Unacast, and Cuebiq released anonymized human mobility data to help combat COVID-19. These novel datasets provide important information about how people move around and how people interact with different types of places, typically called points-of-interest (POIs) [25–29]. Many studies have been conducted based on these human mobility data to understand public compliance with stay-at-home orders [30–32] and the social, economic, and environmental effects of COVID-19, such as decreased restaurant visits and changes in air pollution [33–37]. Among these human mobility data providers, SafeGraph has opened up their data to the research community for free.

In this work, we analyze the alcohol sales data from NIAAA and the human mobility data from SafeGraph to examine the changes in alcohol sales and alcohol outlet visits before and during COVID-19 and their geographic differences. The former dataset allows us to understand the sales changes in spirits, wine, and beer, while the latter enables us to examine the visiting behavior of people to related alcohol outlets, such as liquor stores, bars, wineries, and breweries. The research questions (RQs) that we aim to answer are as follows:

- **RQ1**: Was there indeed a surge in alcohol sales since March 2020? How did alcohol sales vary across different geographic areas and over time?

- **RQ2**: Did people change their visiting behavior to alcohol outlets since March 2020? How did this change vary across different geographic areas and over time?

- **RQ3**: How did the relation between alcohol sales and outlet visits change since March 2020? How did this relation change vary across different geographic areas and over time?

By addressing these three RQs, we advance the understanding of how COVID-19 impacted alcohol sales and people's visiting behavior to alcohol outlets.

## Data and methods

### Data

**Alcohol sales data from NIAAA.** The alcohol sales data were downloaded from the website of NIAAA. It contains monthly alcohol sales information for 16 states in the U.S. from January 2017 to June 2020. The data includes total sales information for spirits, wine, and beer. However, some states do not have data for all three types. For example, the state of Oregon has data for only wine and beer, while the state of Kansas has data for only spirits and beer. Complete beer sales data is available for 11 states, while complete wine and spirits sales data are available for 13 states. Although having its limitations, this alcohol sales dataset provides a great opportunity for empirically examining the changes of alcohol sales since the pandemic began.

**Human mobility data from SafeGraph.** The human mobility data from SafeGraph were collected based on over 45 million anonymized smart mobile devices (mostly smartphones) and over 3.6 million POIs covering the entire United States. A data quality evaluation conducted by SafeGraph [38] showed that this human mobility dataset is statistically representative for the entire U.S. population at the county scale and coarser scales, such as the state scale that we study herein. This human mobility dataset provides highly valuable information about the visiting behavior of people to different POIs including alcohol outlets. Accordingly, we can obtain, for example, the number of times that a liquor store has been visited during a month, allowing us to examine the visiting patterns of people to different alcohol outlets. For privacy protection, this dataset does not provide individual-level trajectories but aggregates the visits of people to POIs and to census blocks. The total file size of this dataset is over 200 GB. This big geospatial dataset provides a unique window for understanding the visiting behavior of people to alcohol outlets before and during COVID-19. The human mobility data cover the time period from January 2018 to the most recent month (the data is usually made available by SafeGraph several days after each month ends).

### Data processing and measures

**Alcohol sales.** The alcohol sales data from NIAAA is organized based on the state, month, and alcohol type. They record the gallons of beverage and ethanol (pure alcohol) sold per month in each state. *Ethanol per capita*, which represents the average amount of pure alcohol sold to an individual, is a main measure provided in this dataset. According to the description of the NIAAA dataset, *ethanol per capita* is calculated using Eq (1):

$$ethanol\ per\ capita = \frac{total\ ethanol\ sold}{total\ population\ age\ 14\ and\ older} \tag{1}$$

The measure of *ethanol per capita* is available for individual states, months, and the three types of alcohol. Since it is a main measure provided by the NIAAA, we use it in this work for quantifying alcohol sales. While the NIAAA data is available since the beginning of 2017, we focus on the data starting from January 2018 in order to make the time range of analysis consistent with the availability of the human mobility data.

**Alcohol outlet visits.**   We derived alcohol outlet visits from the anonymized human mobility data provided by SafeGraph. This dataset provides the number of times a POI is visited during a period. We focus on four types of POIs that have a clear link to alcohol sales: *Beer, Wine, and Liquor Stores* (445310), *Drinking Places* (722410), *Breweries* (312120), and *Wineries* (312130). The names of the four POI types are taken verbatim from the SafeGraph data, which are based on the categories of the North American Industry Classification System (NAICS). The numbers in the parentheses are their corresponding NAICS codes. Note that *Drinking Places* here specifically refer to bars and pubs that sell alcoholic beverages. While people can purchase alcohol from other types of POIs such as grocery stores, a clear link cannot be established between a visit to a grocery store and alcohol purchase. Thus, we chose to focus on these four types of alcohol outlets. We compute the measure *visits per capita* (Eq (2)) to quantify the average number of visits per person to each of the four types of outlets:

$$visits\ per\ capita = \frac{\sum_{i=1}^{n} v_i}{\sum_{j=1}^{m} s_j} \tag{2}$$

where $v_i$ is the number of visits to a particular POI (e.g., a liquor store) in a state during a month, and $\sum_{i=1}^{n} v_i$ is the total number of visits to all POIs of the same type in a state in that month. Variable $s_j$ is the total number of mobile devices in the data whose home locations are within a census block group (CBG; the smallest spatial unit used by SafeGraph), and $\sum_{j=1}^{m} s_j$ is the total number of mobile devices from all CBGs inside a state in that month. Putting these components together, Eq (2) quantifies the average number of visits paid by an individual to a type of alcohol outlets in a given state and a given month. We use Eq (2), i.e., *visits per capita*, as our main measure for alcohol outlet visits, and computed *visits per capita* for each of the four types of alcohol outlets respectively.

## Machine learning

While alcohol outlet visits and alcohol sales are intuitively linked, the pandemic of COVID-19 may prompt people to change their behavior in a variety of ways. For example, one may choose to purchase a large number of bottles in one visit to liquor stores in order to reduce the total number of visits; one may start purchasing alcohol online without having to visit physical stores; or, one may purchase alcohol from the same sources but increase their frequency of visits. It is also possible that one may not largely change their alcohol purchasing behavior, despite other changes in their life during the pandemic. By performing Pearson's correlation analysis between alcohol outlet visits and alcohol sales for individual states in the data, we found a statistically significant correlation between the two: the correlation coefficients are 0.506 ($p<0.001$) for spirits, 0.211 ($p<0.001$) for wine, and 0.414 ($p<0.001$) for beer. Given these statistically significant correlations, we built machine learning models to further examine their relationship and how it has changed since the pandemic began. We trained three different types of models, namely linear regression, random forest, and deep neural network, on pre-COVID data between January 2018 and February 2020, and used the trained models to estimate alcohol sales since March 2020. We then analyzed the difference between the model estimates and the alcohol sales recorded in the NIAAA data. We used random forest and deep neural networks, rather than linear regression alone, because these two machine learning

**Table 1. Four categories of input features for the machine learning models.**

| Category | Input Feature |
|---|---|
| Alcohol outlet visits | Visits per capita to *Beer, Wine, and Liquor Store* |
| | Visits per capita to *Drinking Places* |
| | Visits per capita to *Wineries* |
| | Visits per capita to *Breweries* |
| Time | Month |
| Location | Minimum latitude of the state's geographic boundary |
| | Maximum latitude of the state's geographic boundary |
| | Minimum longitude of the state's geographic boundary |
| | Maximum longitude of the state's geographic boundary |
| State | Dummy variable for each state |

models can effectively capture the complex and nonlinear relationships between the input features and the target variable (e.g., nonstationary trends and periodicity). In the following, we present the input features of the machine learning models, their architectures, and the training process.

**Input features.** We designed four general categories of input features for training the machine learning models (Table 1). The target variable to be estimated is the sales of a type of alcohol (e.g., spirits), as measured by *ethanol per capita*.

The four categories of input features are designed to be general. *Alcohol outlet visits* provide information about the visiting behavior of people to the four types of outlets in a state and during a month. The category *Time* captures the monthly pattern associated with alcohol sales and is represented using the numeric value of each month (i.e., 1, 2, 3, ..., 12). The category *Location* captures the general geographic location of a state by its minimum and maximum latitudes and longitudes, which can indirectly affect alcohol sales (e.g., states with higher latitudes are likely to experience cold winters that could drive alcohol sales). Finally, the category of *State* uses a dummy variable (which is encoded as a one-hot vector) to capture other aspects specific to a state, such as cultures and socioeconomic factors. While these four categories cannot represent all factors that affect alcohol sales, they are general and allow our analysis to focus on the influence of alcohol outlet visits on alcohol sales.

**Machine learning models.** We trained three commonly used machine learning models to fit the relation between the four categories of input features and the target variable of alcohol sales. These three machine learning models are: multiple linear regression (MLR), random forest (RF), and deep neural network (DNN).

- Multiple linear regression: MLR models the relation between multiple input features and the target variable via a linear equation. Specifically, the MLR model used in this work is in the form of Eq (3):

$$y = \theta_0 + \theta_v v + \theta_m m + \theta_l l + \theta_s s + \varepsilon \tag{3}$$

where $\theta_v$, $\theta_m$, $\theta_l$, $\theta_s$ are the regression coefficients for alcohol outlet visits, month, geographic location, and state respectively. Note that each of $\theta_v$, $\theta_l$, $\theta_s$ contains multiple coefficients for the input features in that category (e.g., $\theta_v$ contains four regression coefficients for the visits to the four types of alcohol outlets). We used Python and the scikit-learn library to implement the MLR model.

- Random forest: RF is an ensemble model with many decision trees trained on randomly selected subsets of the training data. The RF model then computes the average of the predictions from the many different decision trees. Compared with MLR, which assumes a linear relation, RF can model the potentially nonlinear relation between the input features and the target variable. We used Python and the scikit-learn library to implement the RF model. The initial RF model tested was based on the default setting of scikit-learn with 100 decision trees. We then performed hyperparameter tuning to identify the optimized decision tree numbers and the number of features to consider at each split.

- Deep neural network: DNNs and other deep learning models have shown outstanding performances in recent years [39]. A DNN uses multiple layers of neurons to learn a complex nonlinear relationship between the input features and the target variable. Here, we built a DNN architecture that has four hidden and fully connected layers with 128, 128, 64, and 32 neurons, respectively (Fig 1). Each neuron uses the nonlinear ReLU activation function, and a dropout rate of 20% is applied to the first three hidden layers to reduce overfitting. The output layer is a single neuron with no activation function that gives the estimated alcohol sales. This model architecture was obtained via a series of experiments in which we tested many other configurations by, e.g., changing the number of neurons per layer, the number of total layers, adding or removing dropout, and adding or removing batch normalization. The DNN model that we present was found to have the best performance among all tested architectures. This DNN model was implemented using Python and the TensorFlow library. Mean square error was used as the loss function for training the model. The model was trained with 200 epochs using the Adam optimizer with its default learning rate of 0.001.

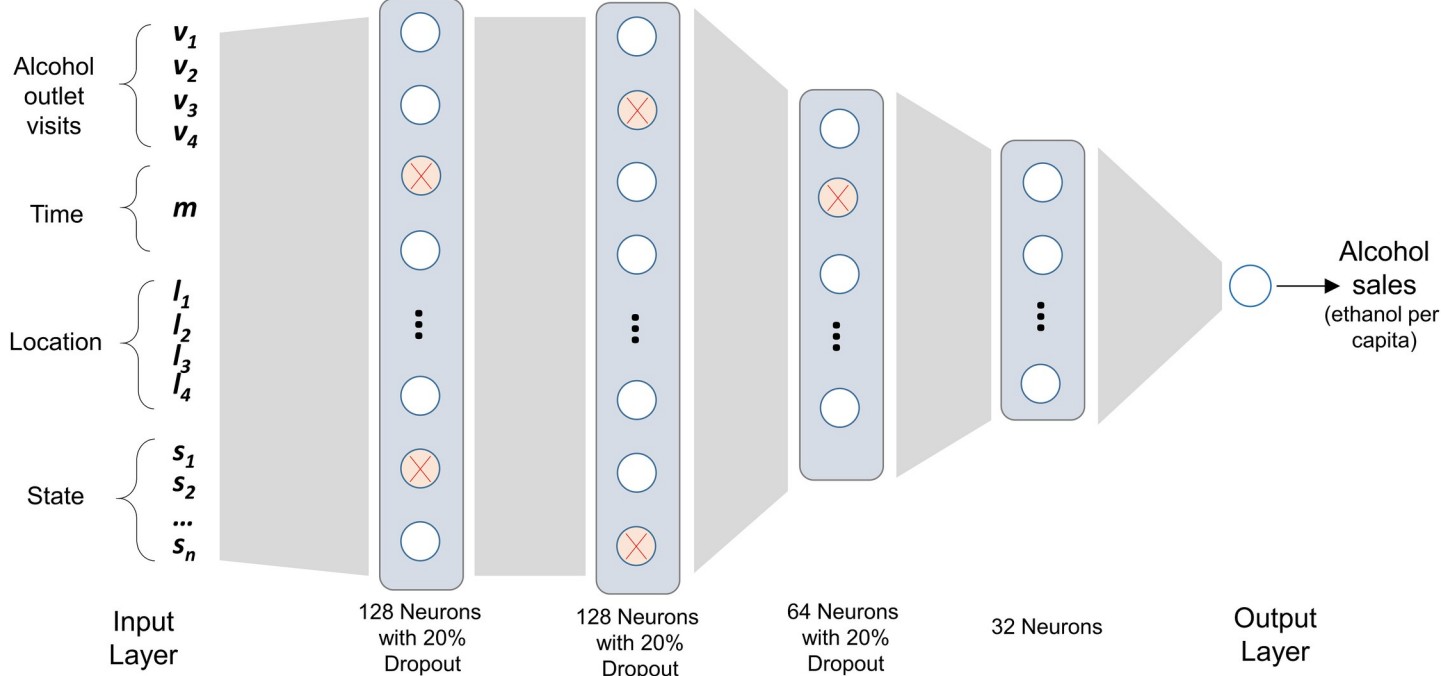

**Fig 1. The architecture of the DNN model for estimating alcohol sales.**

**Model training and evaluation.** All three models use the same input features as shown in Table 1 to estimate alcohol sales. We trained the models using the same training data between January 2018 and February 2020, and performed ten-fold cross-validation to evaluate the trained models. In the ten-fold cross-validation, the training data were randomly divided into ten folds with each fold having 10% of the training data; then, nine folds of the data were used for training and one fold was used for validation. Considering that the models will be applied to predicting future alcohol sales, we also used walk-forward validation which is specifically suitable for time series data [40]. Starting from a model trained using the data in the whole year of 2018, we validated the model using the data in the next three months and walked the model forward month by month with the trained model always predicting alcohol sales in the next three months. The metric we used to evaluate the performance of the trained models is root mean squared error (RMSE) as in Eq (4):

$$RMSE = \sqrt{\frac{1}{m}\sum_{k=1}^{m}(\hat{y_k} - y_k)^2} \tag{4}$$

where $\hat{y_k}$ is the estimated alcohol sales and $y_k$ is the observed alcohol sales in the NIAAA data; $m$ is the total number of data records. We trained the three models for each of the three alcohol types. Therefore, nine models were trained in total.

After the models were trained and evaluated, we applied them to the alcohol outlet visits and other input features between March and June 2020 to make estimates on alcohol sales. We then compared the estimated alcohol sales to the recorded sales in the NIAAA data. If there is a large difference between the model estimate and actual sales, it is likely that people have changed their alcohol purchasing behavior in a way that is no longer captured by the trained model. Note that all three models have alcohol outlet visits as part of their input features; therefore, the trained models should still be able to provide fairly good estimates after March 2020 if people only changed their visiting frequency to alcohol outlets. However, people may change their alcohol purchasing behavior in ways beyond what the trained models can capture. For example, people may purchase a much larger amount of alcohol in a single visit than they normally do, or they may purchase alcohol online without visiting physical stores anymore. Those situations can lead to major deviations between model estimates and recorded alcohol sales.

## Results

### Changes in alcohol sales

The NIAAA data contain alcohol sales for spirits, wine, and beer respectively. We will separately present the sales changes for each of the three types of alcohol.

**Spirits.** Our analysis for spirits sales focused on the 13 states that have complete spirits sales data in the NIAAA dataset: Alaska (AK), Arkansas (AR), Florida (FL), Illinois (IL), Kansas (KS), Kentucky (KY), Louisiana (LA), Massachusetts (MA), Missouri (MO), North Dakota (ND), Texas (TX), Virginia (VA), and Wisconsin (WI). Fig 2(A) shows the total spirits sales for all 13 states from January 2018 to June 2020, and Fig 2(B) shows the percentage changes of total spirits sales (*ethanol per capita*) in March, April, May, and June 2020, compared to the average sales of the same months in 2018 and 2019.

Observe that there was indeed a surge in spirits sales in the months of March, April, and June in 2020. In particular, the spirits sales in March increased by 10.7% compared with the average sales for the same month in 2018 and 2019. Considering that there may exist a natural increase in the consumption of spirits from year to year due to population growth, we also computed the annual change for each month from January 2018 to February 2020, and then

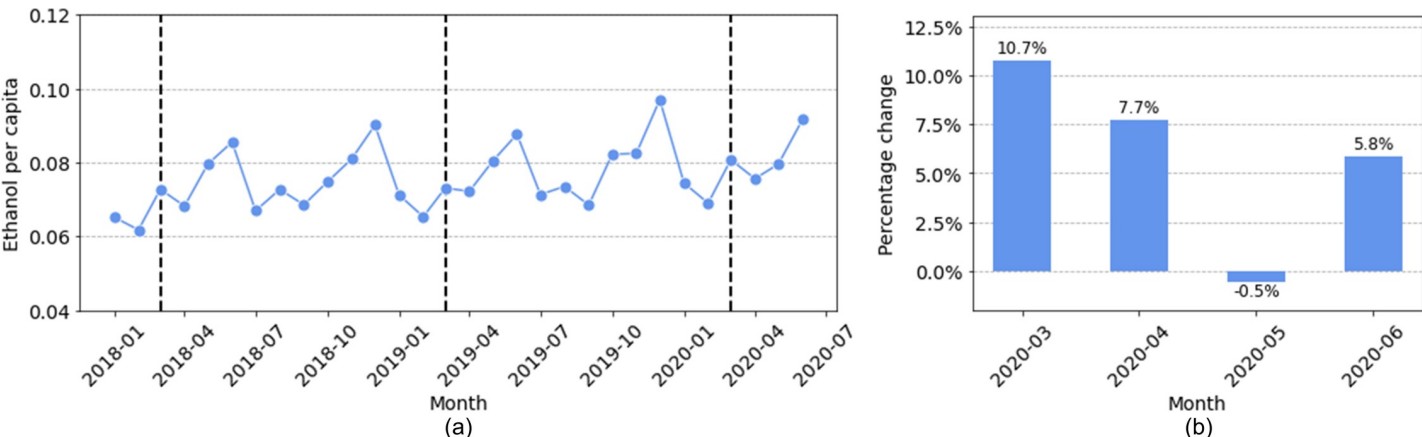

**Fig 2.** Total spirits sales in 13 states: (a) sales in *ethanol per capita* from January 2018 to June 2020 (the dashed lines indicate the month of March in 2018, 2019, and 2020); (b) sales changes in March, April, May, and June 2020 as compared with the average sales in the same months during 2018 and 2019.

summarized their mean and standard deviation. We found a mean increase of 2.8% for spirits sales with a standard deviation of 3.9%. In comparison, the 10.7% increase of spirit sales in March 2020 is approximately two standard deviations larger than the mean, suggesting a significant increase in spirits sales.

Having observed the significant increase in total spirits sales, we examined the sales changes in individual states. We were motivated by the question of whether individual states had sales changes similar to the average, or if there existed large geographic differences across states. The spirits sales in each of the 13 states are shown in Fig 3.

We highlight three major observations. First, observe that the total spirits sales vary largely across different states. Some states, such as AK, ND, and WI, sell about twice the amount of spirits (measured by *ethanol per capita*) as compared to other states, such as AR, TX, and VA. Second, several states (e.g., IL, MA, and ND) show seasonal patterns in their spirits sales, which are not seen in some other states like LA and WI. Third, observe that after the pandemic began, large increases in spirits sales occurred in some states, particularly in March. For example, the spirits sales in AK, MO, and ND in March 2021 are larger than those in the same month in previous years.

To further quantify the spirits sales change of each state in the months of March, April, May, and June, we compared the state-specific sales with the average sales in each state in 2018 and 2019 and visualized them in Fig 4. As can be seen, spirits sales in March 2020 increased in all except two of the states in the data, and four states, in particular, had increases of 20–40%, which are much larger than the average increase of 10.7% of all 13 states in that month. The increase of spirits sales reduced in most states in April and further reduced in May. However, there were increases again in the majority of the states in June 2020. Four states, namely TX, MO, KY, and VA, showed a sustained increase in spirits sales in all four months from March to June 2020. Such increases could be an alarming signal for increased problematic alcohol use in these states.

**Wine.** The NIAAA data contain complete wine sales data for 13 states: Alaska (AK), Arkansas (AR), Florida (FL), Illinois (IL), Kentucky (KY), Louisiana (LA), Massachusetts (MA), Missouri (MO), North Dakota (ND), Oregon (OR), Texas (TX), Virginia (VA), and Wisconsin (WI). Fig 5(A) shows the total wine sales for these 13 states, and Fig 5(B) shows the percentage change of total wine sales in March, April, May, and June 2020 as compared with the average sales of the same months in 2018 and 2019. Overall, we observed a pattern that is similar to the spirits sales; there were increases in wine sales in March, April, and June, but the

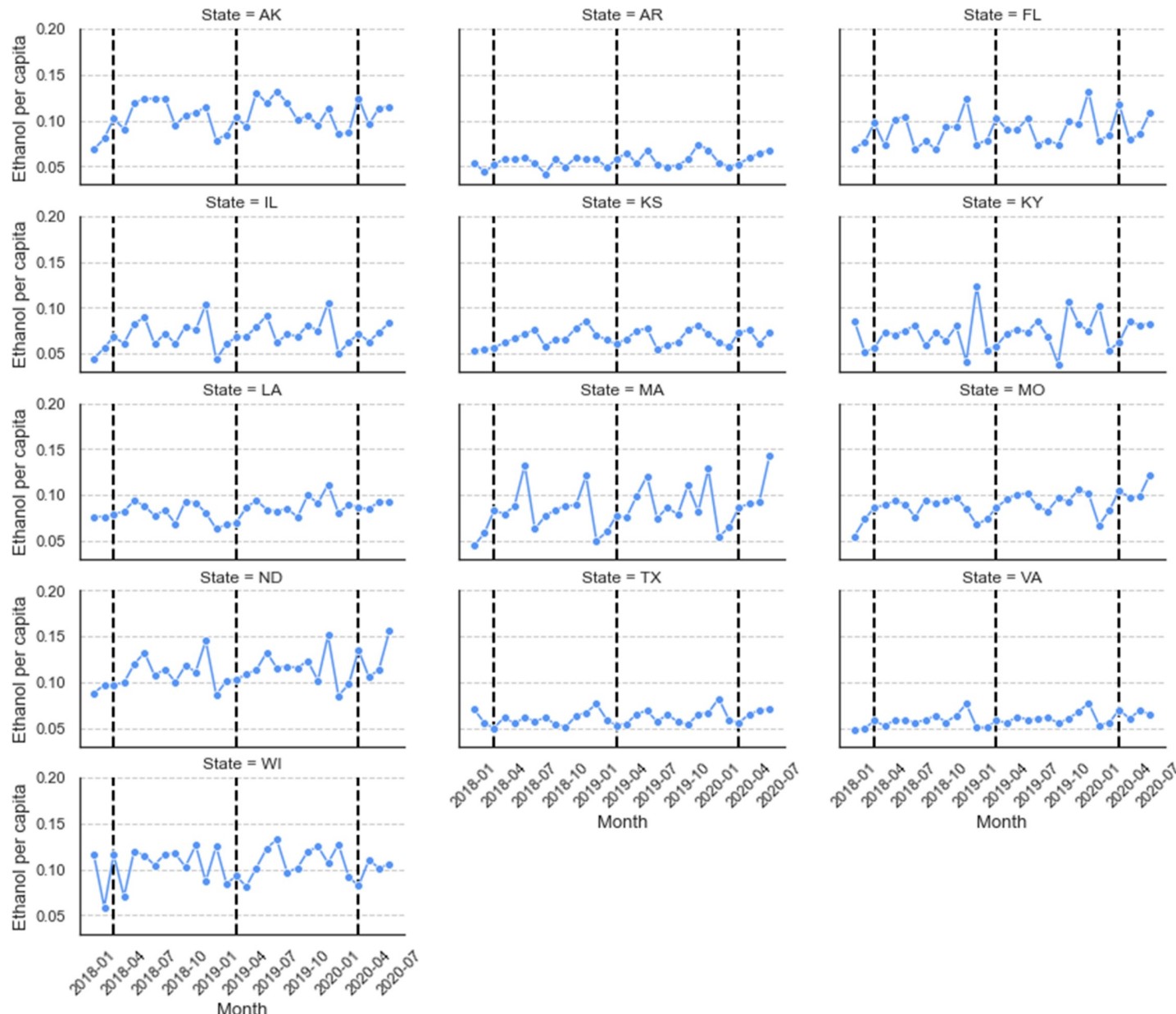

**Fig 3.** Spirits sales in each of the 13 states (the dashed lines indicate the month of March in 2018, 2019, and 2020).

sales decreased in May. The largest sales increase (8.7%) was also observed in March. In comparison, the annual wine sales change for each month between January 2018 and February 2020 had a mean of -0.9% with a standard deviation of 3.2%. The large increase of 8.7% in March 2020 is again two standard deviations above the mean, suggesting a significant increase in wine sales.

We further examined wine sales changes in individual states to study their differences. In S1 Fig of the supplementary material, we plot the wine sales for each month for each state. These results are similar to those that were shown in Fig 3, so we defer them to the supplement. Like the sales of spirits, there exist large differences in the total wine sales across states. For example, the per capita wine sales for some states, such as AK, OR, and MA, are about twice

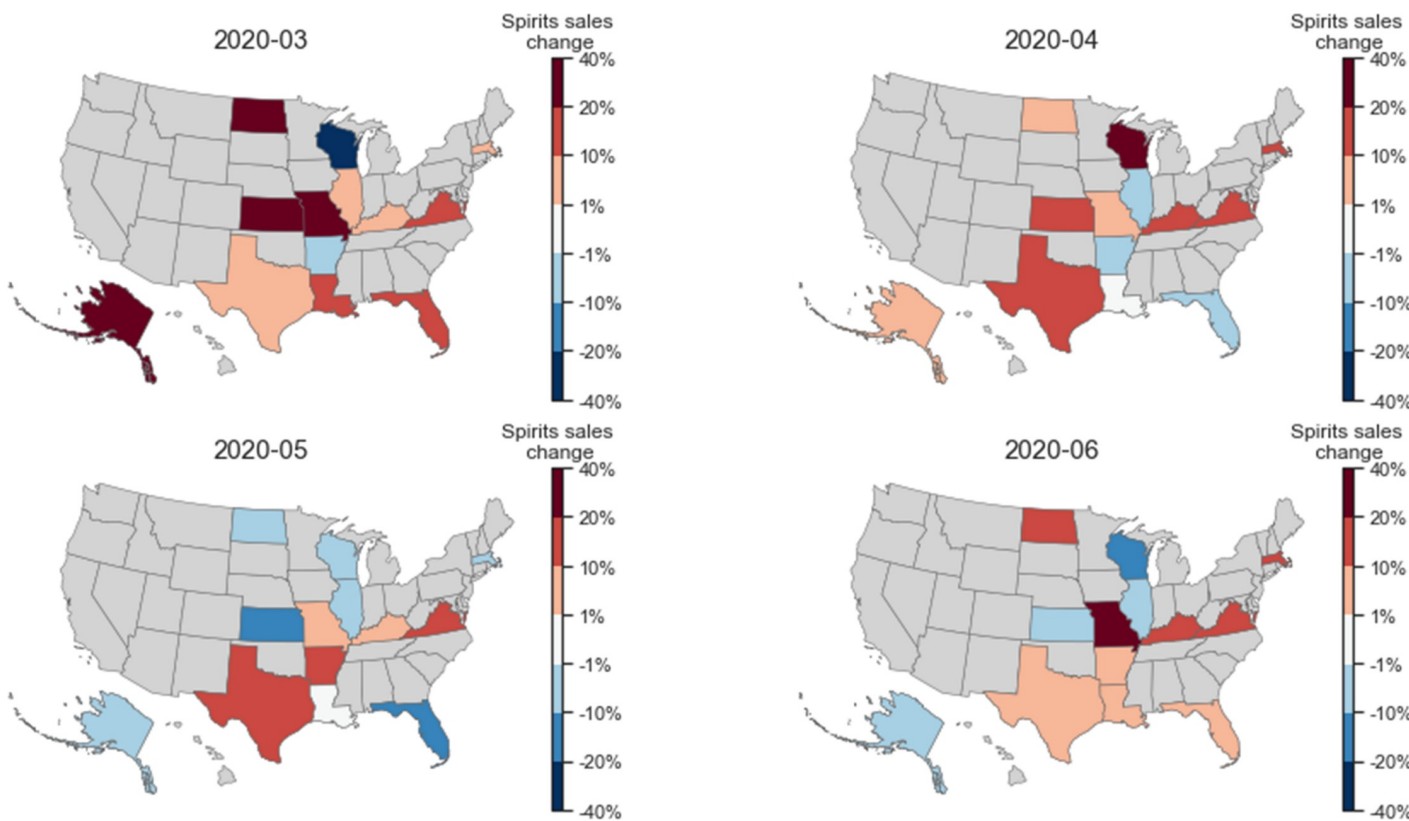

**Fig 4. Geographic differences for how the per capita sales of spirits changed during the pandemic.** The percentage change for each state is compared to the average sales during the same months in 2018 and 2019. Red colors indicate an increase in spirits sales, while blue colors indicate a decrease in sales. The darker the colors, the larger the changes.

that of other states, such as AR, KY, and TX. Seasonal patterns also exist in the wine sales of some states (e.g., MA and ND) but not others (e.g., TX and WI).

We quantified the changes in wine sales in individual states from March to June 2020 by comparing them to the average sales in the same months of 2018 and 2019, as shown in Fig 6.

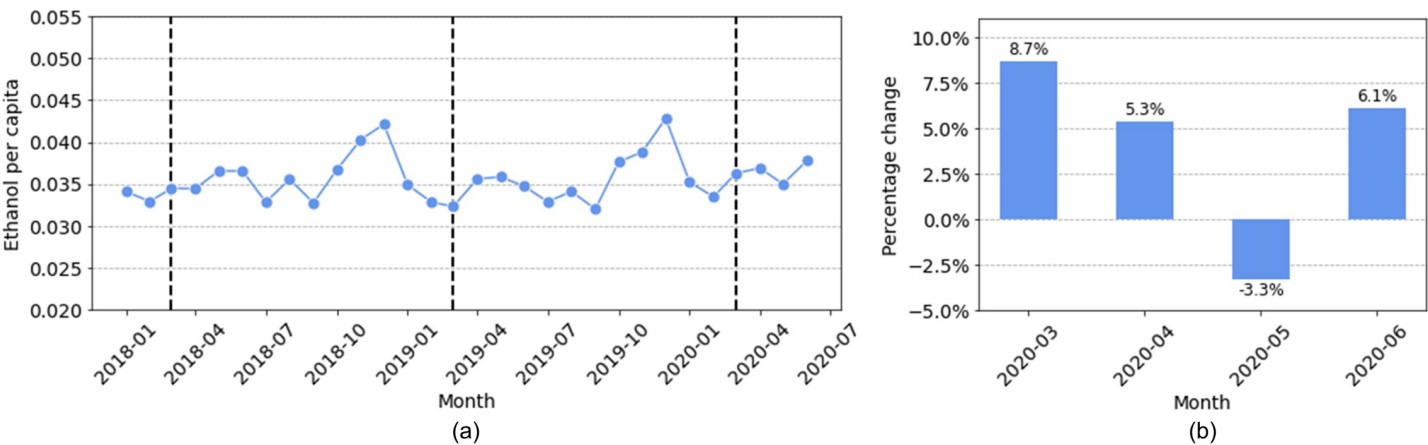

**Fig 5.** Total wine sales in thirteen states: (a) sales in *ethanol per capita* from January 2018 to June 2020 (the dashed lines indicate the month of March in 2018, 2019, and 2020); (b) sales changes in March, April, May, and June 2020 as compared with the average sales in the same months during 2018 and 2019.

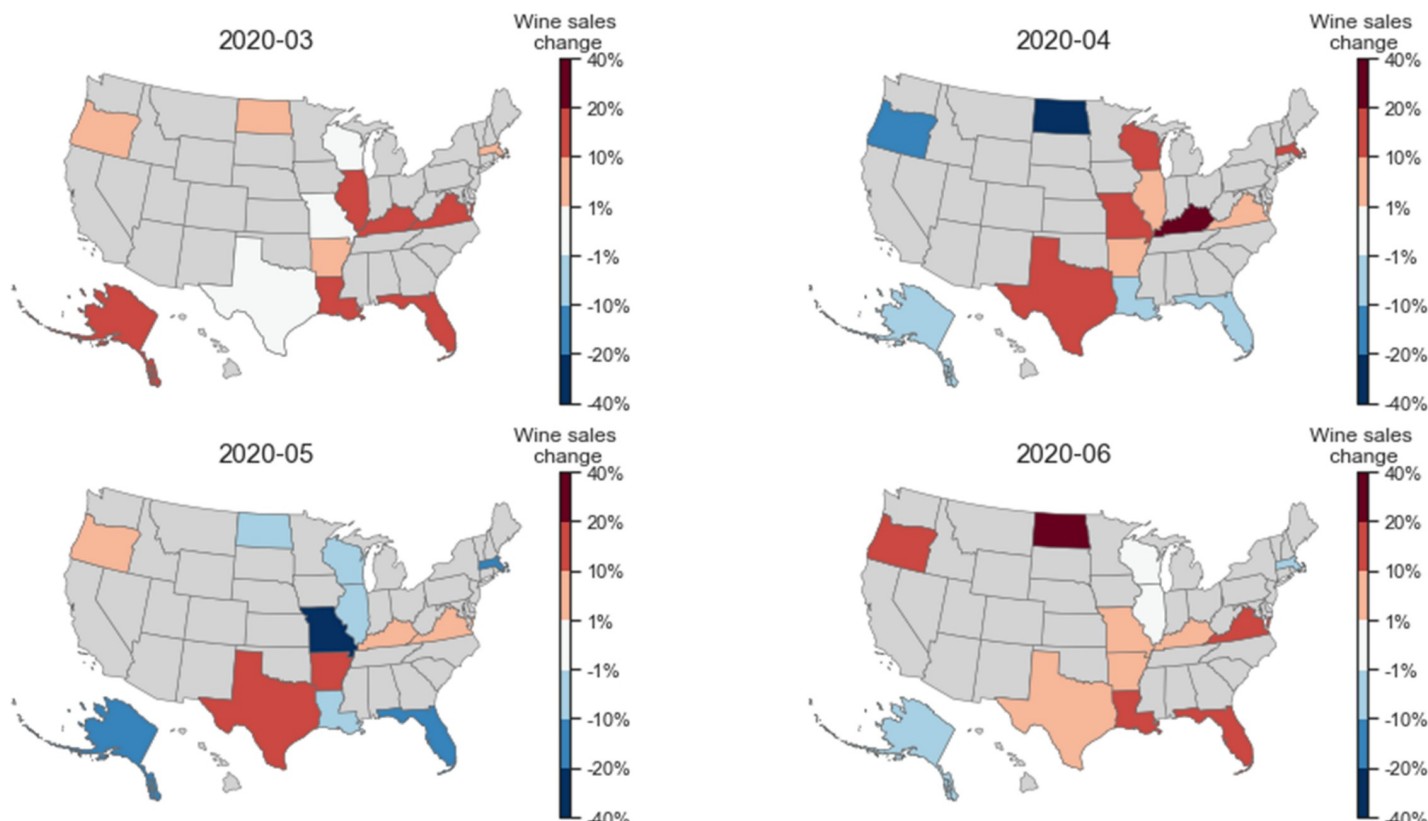

**Fig 6. Geographic differences for how the per capita sales of wine changed during the pandemic.** The percentage change for each state is compared to the average sales during the same months in 2018 and 2019.

The sales of wine in March 2020 markedly increased in all states except for three, which maintained roughly the same level of sales (i.e., between -1% to +1%). None of the thirteen states had a decrease in wine sales in March. Some states decreased their wine purchase in April, and more states decreased their wine sales in May. However, nine of the states increased their wine sales again in June 2020. Three states, namely AR, KY, and VA, showed a sustained increase in wine sales in all four months, while the state of TX showed an increase in wine sales in April, May, and June (and similar sales in March).

**Beer.** The NIAAA data contains complete beer sales for 11 states: Alaska (AK), Arkansas (AR), Florida (FL), Illinois (IL), Kansas (KS), Kentucky (KY), Massachusetts (MA), Missouri (MO), North Dakota (ND), Oregon (OR), and Texas (TX). Fig 7(A) shows the total beer sales for all eleven states, and Fig 7(B) shows the percentage change of beer sales in March, April, May, and June 2020 as compared to the average sales for the same months in the previous two years. The pattern for beer sales is strikingly different from that for spirits and wine: beer sales decreased in March, April, and May, and only slightly increased in June. The largest decrease occurred in March 2020 with a decrease of 7.1%. For comparison, we computed the annual change of beer sales for each month between January 2018 and February 2020, and observed a mean of -1.2% with a standard deviation of 3.4%. The decrease of 7.1% in beer sales in March 2020 is roughly 1.7 standard deviations below the mean, suggesting a large decrease in beer sales after the pandemic began.

We further looked into how beer sales changed in individual states. In S2 Fig, we plot the per capita beer sales per month for individual states. In Fig 8 below, we depict the changes in

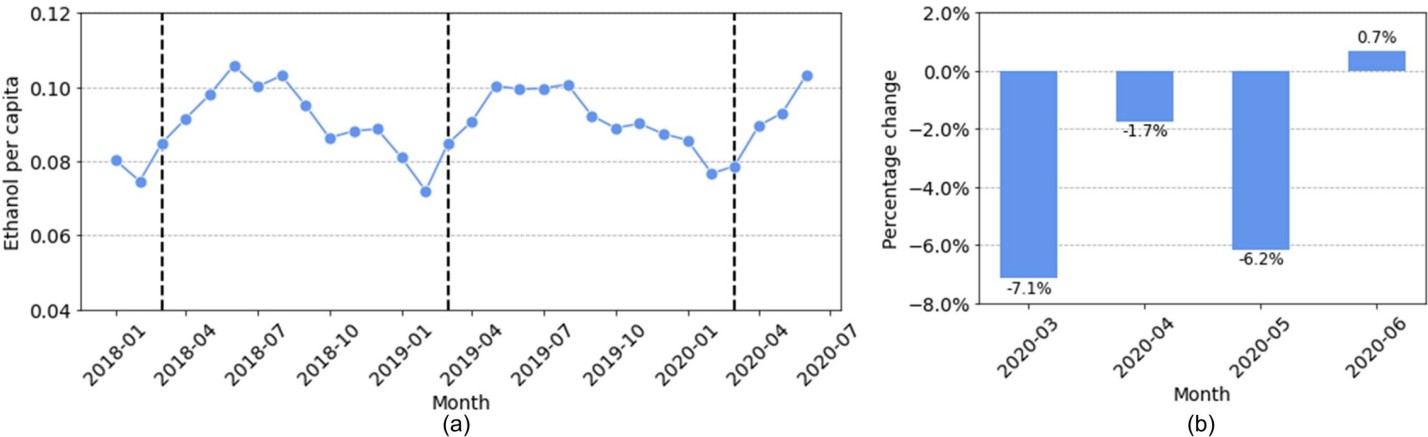

**Fig 7.** Total beer sales in eleven states: (a) sales from January 2018 to June 2020 (the dashed lines indicate the month of March in 2018, 2019, and 2020); (b) sales changes in 2020 compared with the average sales in the same month of 2018 and 2019.

per capita beer sales in individual states. While beer sales decreased in most states in March, April, May, and June, they did increase for a few states, such as KS, AR, and TX, suggesting geographic differences.

**Summary.** In this section, we examined the monthly alcohol sales data from NIAAA in order to answer RQ1: *Was there indeed a surge in alcohol sales since March 2020? How did alcohol sales vary across different geographic areas and over time?* Our analysis showed that there were significant increases in the sales of spirits and wine since March 2020, but the sales of beer decreased during the same period. In addition, geographic differences exist across states: some states showed a sales increase as large as 20–40% in spirits and wine sales, while other states showed only a mild increase or even decrease in sales. Finally, three states, namely TX, KY, and VA, showed sustained increases in their sales of both spirits and wine in March, April, May, and June, which can be alarming signals for problematic alcohol use.

## Changes in alcohol outlet visits

We analyzed the visits of people to four types of alcohol outlets using the human mobility data from SafeGraph in order to understand whether and how the visiting behavior of people to these alcohol outlets changed since the stay-at-home orders began in March 2020. We focused our attention on the same sixteen states that are contained in the NIAAA dataset and on the time window from January 2018 to June 2020. Fig 9(A) shows the *visits per capita* to each of the four types of alcohol outlets, which we average across the sixteen states. Fig 9(B) shows the percentage changes of visits to these alcohol outlets during the months of March, April, May, and June 2020. Similar to our study of alcohol sales, the percentage changes are computed relative to the average value for the same months in 2018 and 2019. (Note that we divide the visits to drinking places by 2 so that all curves can be depicted with a similar scale.)

As one might expect, the per capita visits to three types of alcohol outlets, namely drinking places, breweries, and wineries, largely decreased in response to COVID-19. In particular, we observed a dramatic decrease in the visits to drinking places (e.g., bars and pubs) in March and April during the stay-at-home orders. This is most likely due to the shutdown of these drinking places. Similar decreases were observed in the visits to breweries and wineries. The only exception among the four types of alcohol outlets is liquor stores. While visits to liquor stores slightly decreased in March, they increased in April, May, and June. In particular, we find that the visits to liquor stores increased 21.2% in May 2020. For comparison, we computed the annual

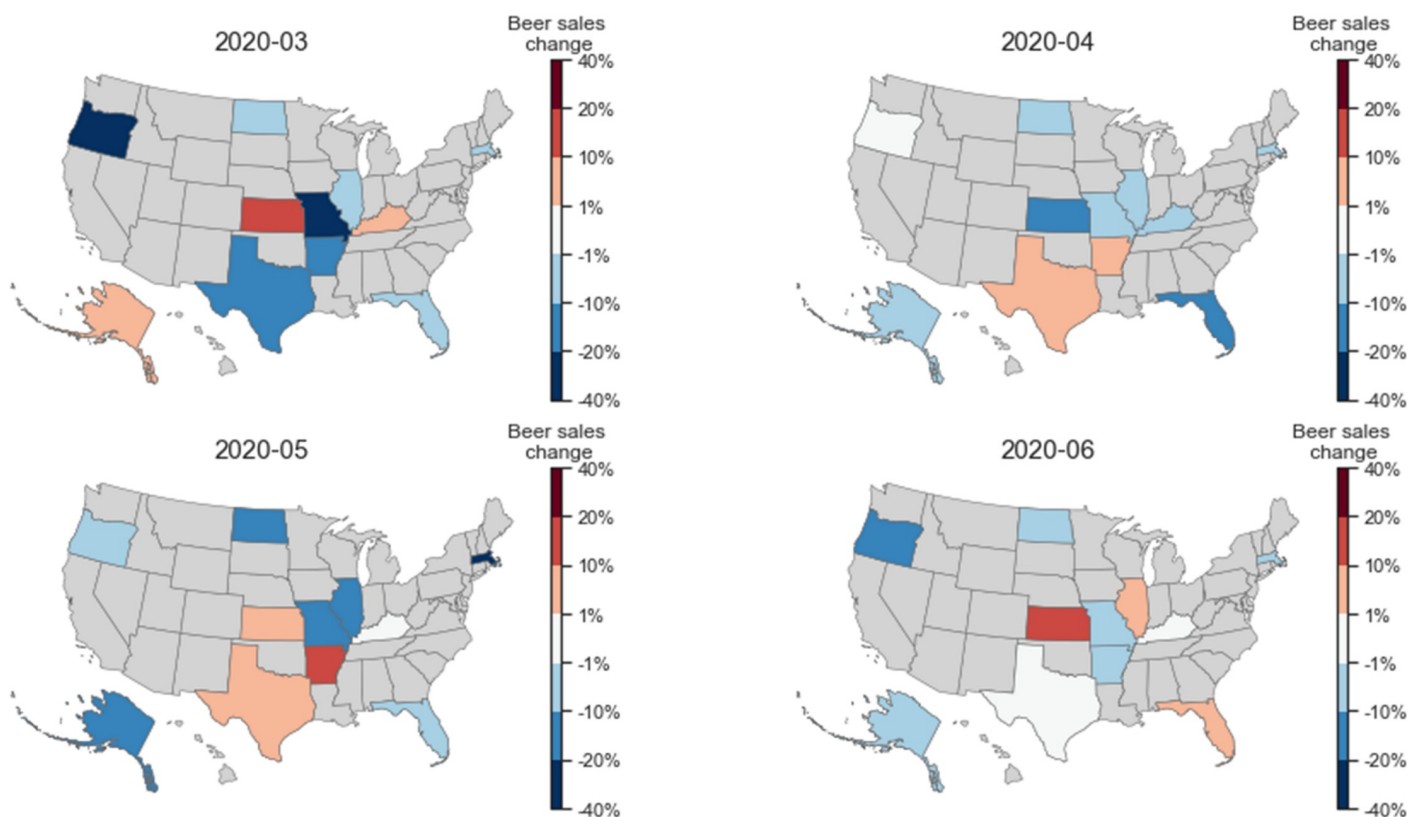

**Fig 8. Geographic differences for how the per capita sales of beer changed during the pandemic.** The percentage change for each state is compared to the average sales during the same months in 2018 and 2019.

change in visits to liquor stores for each month between January 2018 and February 2020, finding a mean increase of 13.8% with a standard deviation of 13.9%. The increase of 21.2% for liquor store visits in May is therefore a moderate increase compared with the pre-COVID time.

We further examined alcohol outlet visits in each of the 16 individual states, and the results are shown in Fig 10. Note that similar to Fig 9(A), we divided the visits to drinking places by 2 so that all curves are visualized within a similar scale. We highlight two main observations. First, different states can have very different patterns in their visits to the four types of alcohol outlets. For example, people in CT, MA, and ND visit liquor stores much more frequently than people in FL, LA, and WI, who visit bars and pubs more frequently than people in some other states. Second, after the pandemic began, there were large decreases in the visits to drinking places across all 16 states, but the visits of people to liquor stores vary across the states. People in states including FL, LA, IL, and TX maintained similar visiting frequencies after the pandemic began, while people in states including AR, KS, CT, and ND increased their visits to liquor stores.

Next, we examined how the visits to liquor stores changed for each state after the pandemic began. In Fig 11, we illustrate for each state the percentage change in per capita visits in March, April, May, and June, which we computed relative to the average value for the same months in 2018 and 2019. As can be seen, people in 14 of the 16 states (except AR and TX) decreased their visits to liquor stores in March 2020 but later increased their visits in April, May, and June. In particular, all 16 states increased their per capita visits to liquor stores in May, and the largest increases of 40–60% are observed in KS and AR. An alarming signal

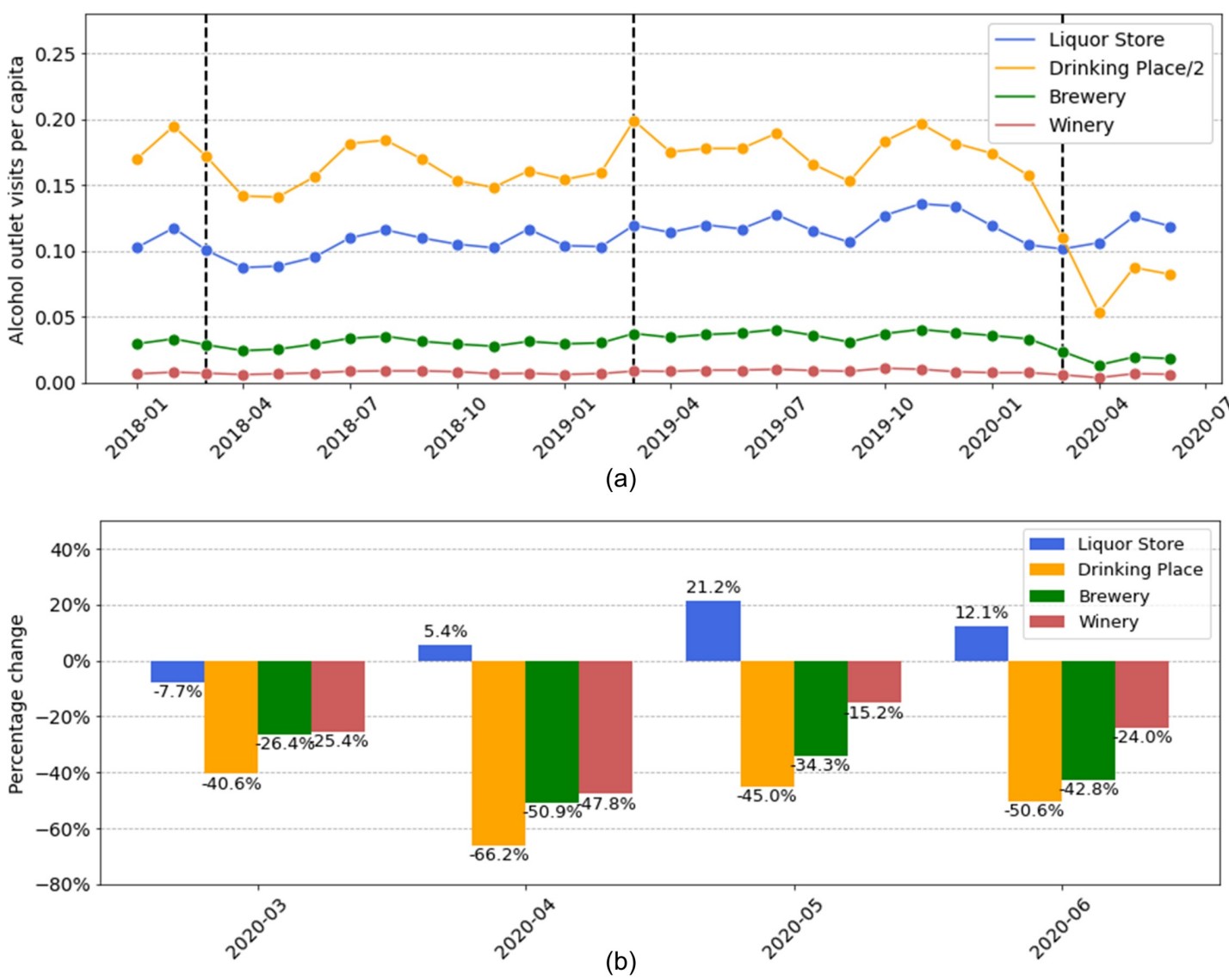

**Fig 9. Visitation behavior to the four types of alcohol outlets in the U.S. during the first months of COVID-19.** (a) Time series describing the per capita visits for each month from January 2018 to June 2020 (which is averaged across 16 states). The dashed lines indicate the month of March in 2018, 2019, and 2020. Note that visits to drinking places are divided by 2 so that all curves can be shown within a similar scale for clear visualization. (b) For the months of March, April, May, and June 2020, we depict the changes to per capita visits. For each month, the percentage change was computed relative to the average value for the same month of 2018 and 2019.

shows up in the state of AR which had constant and large increases in the visits of people to liquor stores.

We also analyzed the changes in visits to the other three types of alcohol outlets, i.e., drinking places, breweries, and wineries, in each individual state, and the results are shown in S3–S5 Figs. We observe that the visits to these other alcohol outlets mostly decreased in individual states (including the state of AR), and we only observe slight increases in visits in KS and MO for wineries in May and June. For visits to drinking places, we observe large decreases of 60–80% in April for 13 out of the 16 states.

**Summary.** In this section, we examined the human mobility data from SafeGraph in order to answer RQ2: *Did people change their visiting behavior to alcohol outlets since March*

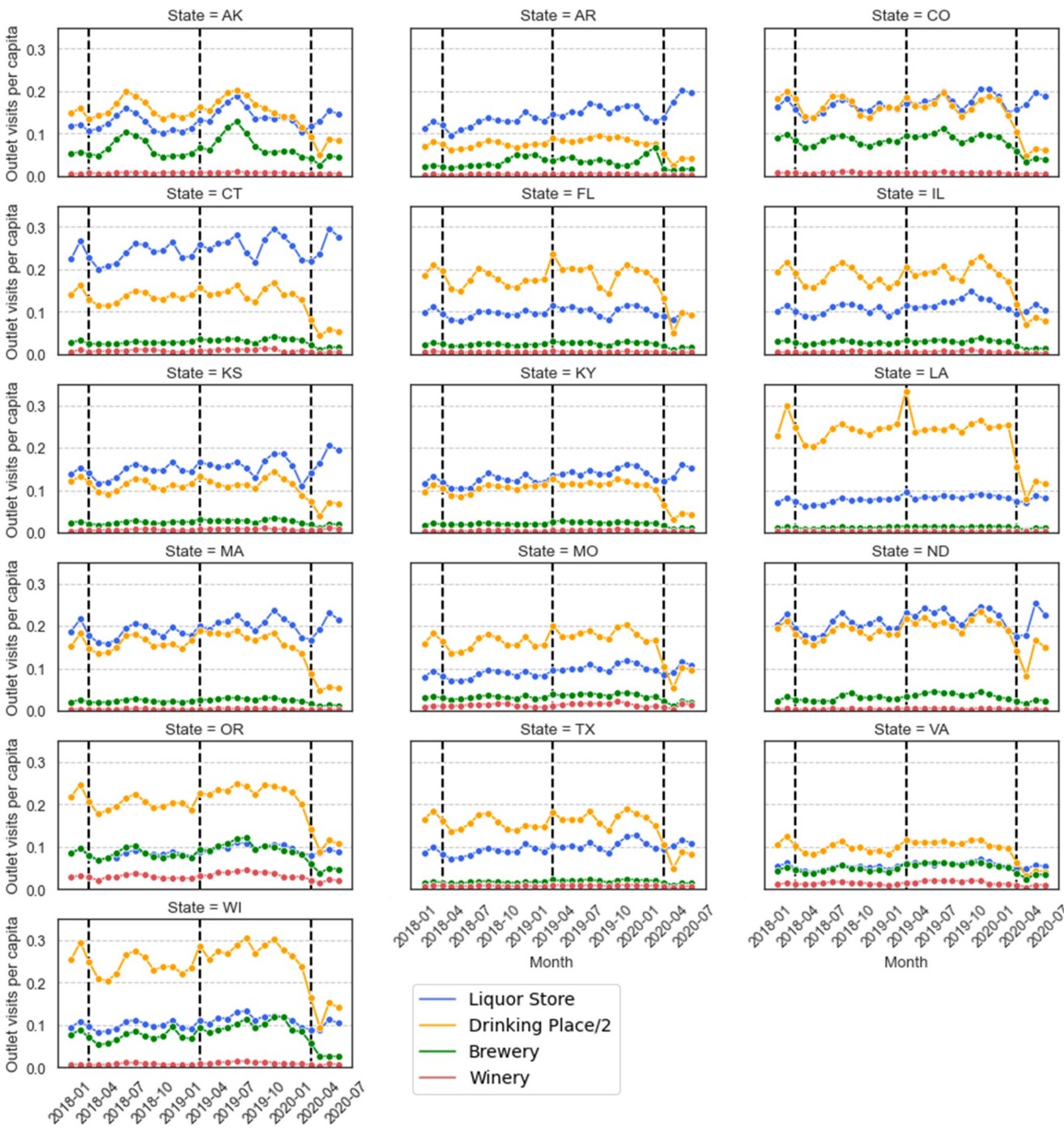

**Fig 10. Visits to the four types of alcohol outlets in each of the 16 states.** The vertical dashed lines indicate the month of March in 2018, 2019, and 2020.

*2020*? *How did this change vary across different geographic areas and over time*? Our analysis showed that people in the 16 states that we studied largely reduced their visits to three types of alcohol outlets: drinking places, breweries, and wineries. In contrast, liquor stores, which remained open in most states [4], received increased visits during the lockdowns and particularly in the month of April 2020 (with an increase of 21.2% based on the average of the 16

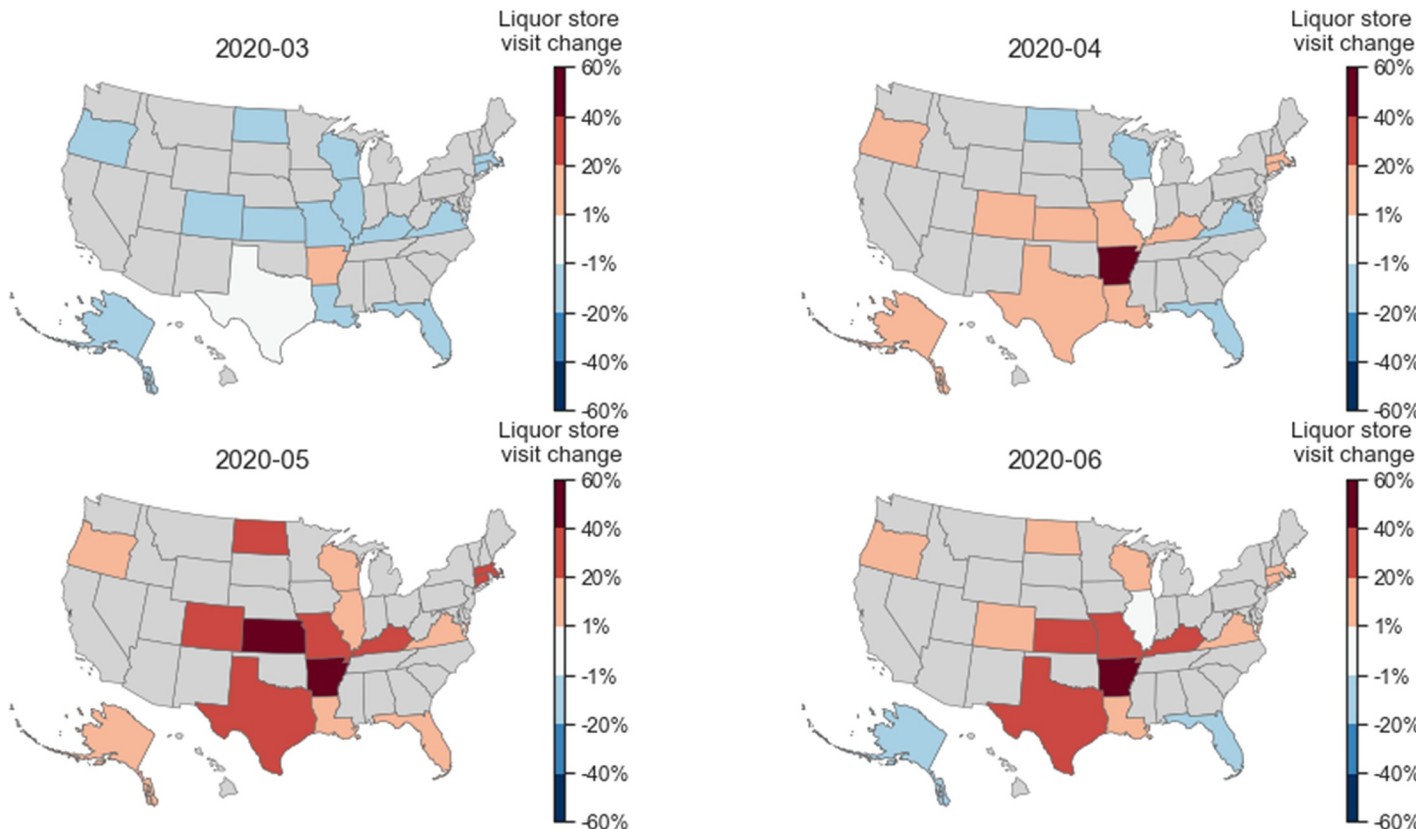

**Fig 11. Geographic differences for how the per capita visits to liquor stores changed in response to COVID-19.** For each month, the percentage change to visits was computed relative to the average for the same month in 2018 and 2019. Red colors indicate an increase in sales, while blue colors indicate a decrease in sales. The darker the colors, the larger the changes.

states). Our analysis also revealed substantial geographic differences across these states regarding the visits to liquor stores. Some states such as AR and KS showed alarming increases of 40–60%, whereas the visits to liquor stores changed very little for other states such as WI and FL. However, there were large (over 60%) decreases across the 16 states for the visits to drinking places, wineries, and brewers since the pandemic began and particularly during the stay-at-home orders in April.

## Changes in the relation between alcohol outlet visits and alcohol sales

Given the natural link between alcohol outlet visits and alcohol sales, we used machine learning models to examine their complex relation and how it changed after COVID-19 began. As described in Section 2, we first trained and evaluated three types of machine learning models, namely MLR, RF, and DNN, using the pre-COVID (i.e., between January 2018 and February 2020) sales and visits data. We then used the best model to estimate alcohol sales since March 2020 and compared the model estimations with the recorded alcohol sales from NIAAA.

**Model performances.** For MLR, we first performed multicollinearity tests by calculating the variance inflation factor (VIF) for the numeric independent variables. Table 2 shows the VIF values for three MLR models in which we gradually removed the variable with the highest VIF value. The tests identified collinearity between minimum and maximum latitudes and longitudes, which can be expected. By removing maximum latitude and then maximum

**Table 2. VIF values of the variables in three MLR models.**

| Input Feature | Primary Model | Reduced Model 1 | Reduced Model 2 |
|---|---|---|---|
| Visits per capita to *Beer, Wine, and Liquor Store* | 2.110 | 2.108 | 1.975 |
| Visits per capita to *Drinking Places* | 1.365 | 1.191 | 1.157 |
| Visits per capita to *Wineries* | 1.835 | 1.777 | 1.727 |
| Visits per capita to *Breweries* | 2.467 | 2.448 | 2.405 |
| Month | 1.117 | 1.112 | 1.106 |
| Minimum latitude of the state's geographic boundary | 23.906* | 3.247 | 2.937 |
| Maximum latitude of the state's geographic boundary | 64.116* | - | - |
| Minimum longitude of the state's geographic boundary | 12.109* | 5.377* | 1.504 |
| Maximum longitude of the state's geographic boundary | 9.956* | 6.542* | - |

*VIF value exceeds 5.

longitude, we obtained *Reduced Model 2* whose VIF values are all smaller than the typically cut-off value 5. However, the RMSE values of the three models are the same. As discussed in the literature, multicollinearity does not affect prediction in general [41], although it affects the obtained coefficients and *p* values. In the following, we used the *Primary Model* since it has the same prediction capability as the reduced models while keeping the input features the same as those used by the RF and DNN models.

The performances of the three models in estimating alcohol sales are shown in Fig 12, with the first row showing the result of ten-fold cross-validation and the second row showing the result of walk-forward validation. RMSE was calculated based on the predicted alcohol sales and the recorded sales in the NIAAA data. The two sets of validation experiments showed similar results: all three models provided reasonable accuracies, while the RF model had the best performance among the three, as demonstrated by its lowest RMSE. The RF model also performed better than the more complicated DNN model. A possible explanation is that this prediction task is based on a small training dataset (there are 338 data records in total, with each data record representing a state in a month). Note that the original human mobility data from SafeGraph is large, but the machine learning models were trained on the data aggregated to state and month levels. When given a small dataset, a simpler and more traditional machine learning model like random forest can be better tuned than a more complicated model like a deep neural network. We also tested the three models with the sales of wine and beer and found similar results in that the RF model consistently outperformed the other two models. Thus, we selected the RF model for estimating alcohol sales.

In our selection of alcohol outlet types, we excluded grocery stores because the link between a grocery store visit and alcohol purchase is unclear. Nevertheless, people can purchase alcohol from grocery stores that remained open during the shutdown period. Thus, we further examined the effect of including grocery store visits as an additional predictor for alcohol sales. We used the RF model for this examination which showed the best performance among the three models. Table 3 summarizes the RMSE values of the RF models which included or not included grocery store visits as one of their input features. The result shows that there is almost no change in the performance of the model when grocery store visits are included. In fact, the RMSE values became slightly worse for wine and beer sales estimations when grocery store visits were included. The experiment result suggests that grocery store visits may be too noisy to effectively contribute to alcohol sales estimation. While people can indeed purchase alcohol from grocery stores, there also exist many grocery store visits which do not involve alcohol purchase.

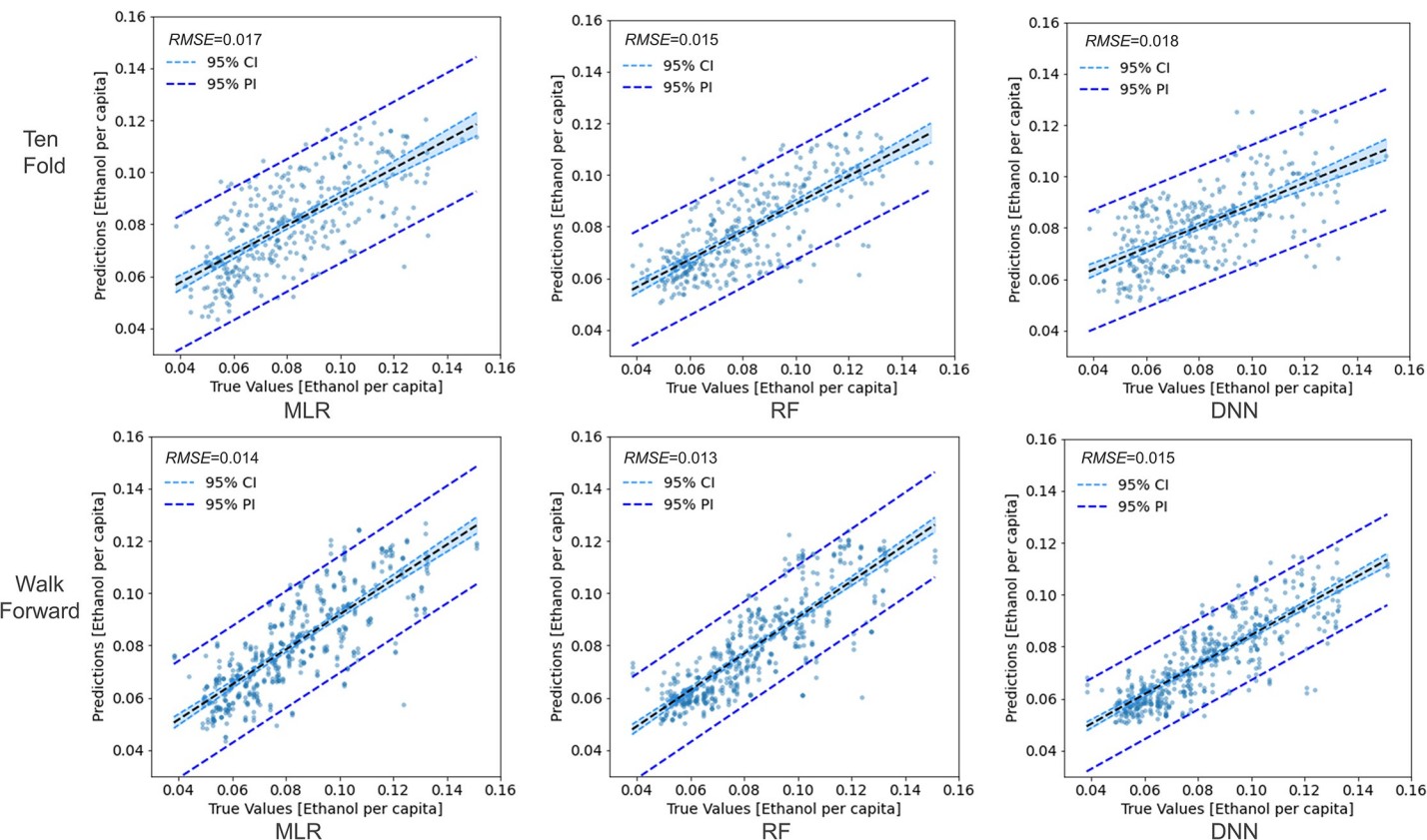

**Fig 12. Performances of the three machine learning models based on ten-fold cross-validation and walk-forward validation: Multiple linear regression (MLR), random forest (RF), and a deep neural network (DNN).**

Next, we tune the hyperparameters of the RF model to identify the optimized parameter configuration. We performed hyperparameter tuning for spirits, wine, and beer respectively. Learning from the literature [31], we focused on tuning two major hyperparameters, namely $n_{tree}$, which controls the number of decision trees, and $m_{try}$, which controls the number of features to consider at each split in a tree. We performed grid search to identify the best hyperparameters, and the search space for $n_{tree}$ is set to [10, 200] with an interval of 10, and the search space for $m_{try}$ is $\{S, \sqrt{S}, log_2 S, S/2, S/3\}$ where $S$ is the number of input features. We limited the search space for the number of trees to 200 because the dataset is relatively small. The identified optimized hyperparameters were then used in the final RF models for alcohol sales estimation.

**Comparison between RF model estimates and recorded alcohol sales.** We used the trained RF model to estimate spirits sales since March 2020, and the results are shown in Fig 13. As can be seen, the RF model did a fine job in capturing the general fluctuations of spirits sales before the pandemic. Since March 2020, however, some large deviations between model estimates and recorded sales were observed in individual states. Particularly, in AK, FL, MO,

**Table 3. RMSE values of including and not including grocery store visits for alcohol sales prediction based on the RF model.**

|  | Spirits | Wine | Beer |
| --- | --- | --- | --- |
| *Including* Grocery Store Visits | 0.01280 | 0.00548 | 0.01201 |
| *Not Including* Grocery Store Visits | 0.01288 | 0.00543 | 0.01174 |

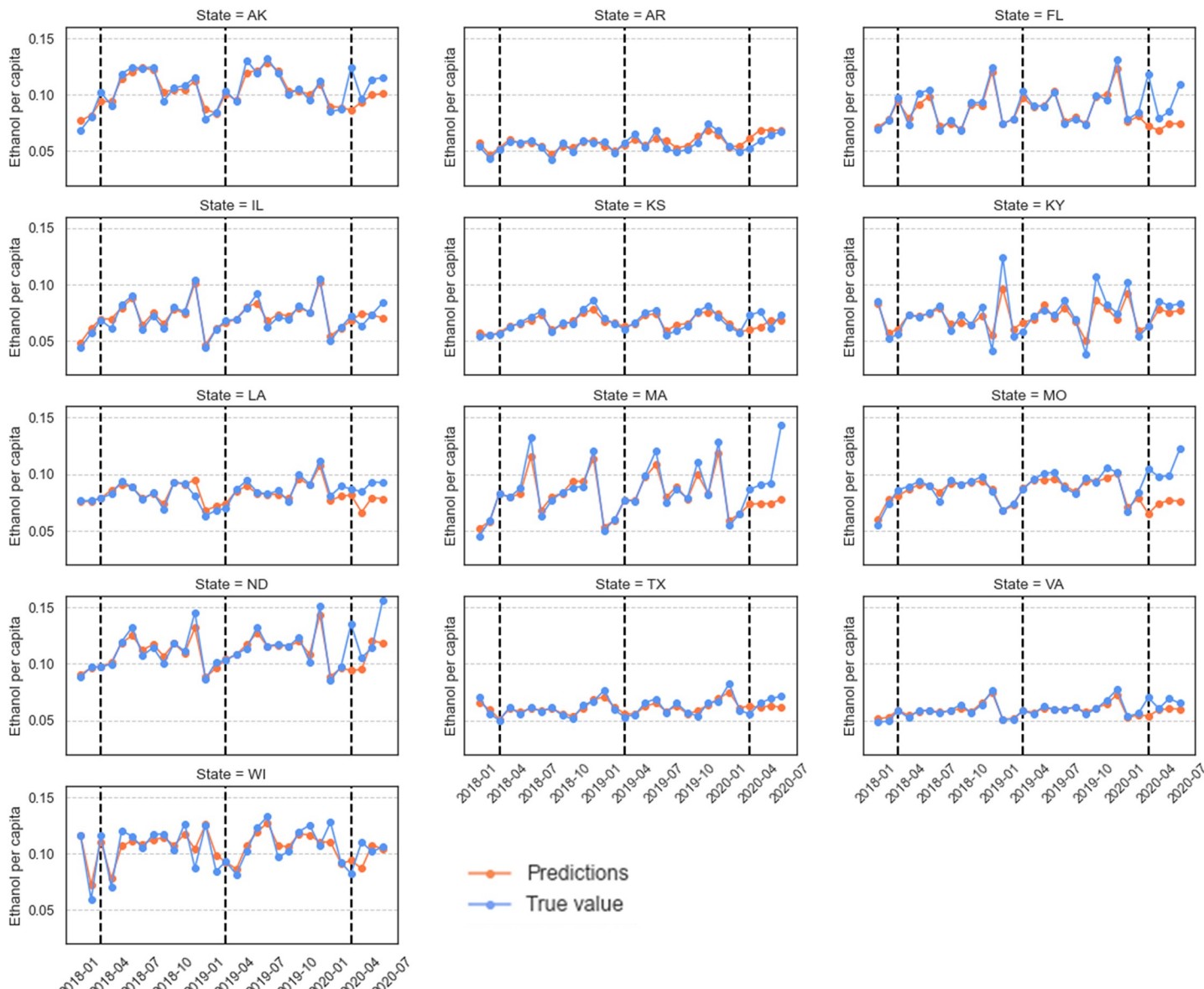

**Fig 13. Model estimates and the recorded per capita sales of spirits in individual states.** The vertical dashed lines indicate the month of March in 2018, 2019, and 2020. The model was trained on data before March 2020 and was then used to predict the sales for the months following (and including) March 2020.

and ND there are major peaks in the sales of spirits in March that far exceed the model estimates. These large differences between the observed and predicted sales suggest that people in these states likely changed their spirits purchasing behavior in a way that is not captured by the trained model. Moreover, since the model predictions utilize the data describing visits to the different alcohol outlets before and during the pandemic, these deviations suggest that people may have changed other aspects of their alcohol purchasing behavior. For example, they may be purchasing larger amounts of spirits during each visit (i.e., panic buying or hoarding), or they may purchase spirits online without having to visit stores. In contrast, in states like AR, IL, KY, and TX the differences between the predicted and recorded spirit sales are relatively small, which suggests that people in these states likely changed their alcohol purchasing

behavior in a way that is captured by the model (e.g., they may have changed their frequency of visits to liquor stores, but they maintained a similar purchasing rate per visit).

We also compared the model estimates and the actual sales for wine and beer, and we provide the results in the supplementary materials. For wine sales, we observed four states (IL, MA, MO, and TX) whose model predictions have relatively large differences from the recorded wine sales since the pandemic began. The model estimates of other states remained close to the recorded sales. Interestingly, the states of IL and TX had estimated spirits sales that were close to the recorded sales but had estimated wine sales that were quite different from the recorded sales. This result suggests that people may change their purchasing behavior for one type of alcohol but maintain their behavior for another type in the face of a pandemic. For beer sales, the model estimates since March 2020 in most states are fairly close to the recorded sales (see S7 Fig). Overall, the results suggest that the purchasing behavior of people on beer changed in a way that is still largely captured by the trained model (e.g., decreased beer sales is linked to the decreased visits of people to bars, pubs, and breweries).

**Summary.** In this section, we examined the human mobility data and alcohol sales data together in order to answer RQ3: *How did the relation between alcohol sales and outlet visits change since March 2020? How did this relation change vary across different geographic areas and over time?* Given the natural link between alcohol sales and outlet visits, we built three types of machine learning models to examine the relation between these two types of data and how that relation changed in response to COVID-19. We trained models using the pre-COVID data and then compared model estimates with the recorded alcohol sales since March 2020. Based on the optimized RF models that achieved the best performance, our result showed that the relation between alcohol sales and outlet visits might have changed in a variety of ways, some of which are captured by our trained model (e.g., increasing visiting frequency to liquor stores), while others are not (e.g., purchasing a large amount of alcohol per visit or purchasing alcohol online). Our results also showed that people likely changed their alcohol purchasing behavior differently with regard to spirits, wine, and beer. In addition, geographic differences were observed regarding how the relation between alcohol sales and outlet visits changed due to the pandemic. States including AK, FL, MO, and ND had model estimates that were much lower than the recorded spirit sales in March 2020, while some other states, such as AR, IL, KY, and TX, had model estimates that were close to the recorded spirits sales even after the pandemic began. Further research is necessary to better understand how exactly people changed their alcohol purchasing behavior in these states.

## Discussion

### Feature importance

The use of the RF model allows us to examine the relative importance of different input features for alcohol sales estimation. The importance of a feature is computed based on its contribution to impurity reduction in the nodes of a tree and then averaged over the trees in the random forest. The importance values output by the RF model are normalized to the range of [0, 1] and sum up to 1. Thus, they suggest the relative importance of an input feature in helping the model make predictions compared with other input features. Fig 14 shows the importance of the numeric features for estimating the sales of spirits, wine, and beer respectively. As can be seen, three spatial and temporal features, i.e., maximum latitude, minimum longitude, and month, play highly important roles in helping the model estimate alcohol sales. We will discuss this further in the next subsection. While the relative importance of other features varies across alcohol types, visits to drinking places are an important feature for predicting the sales of spirits, wine, and beer. In addition, visits to liquor stores also have moderate importance across all

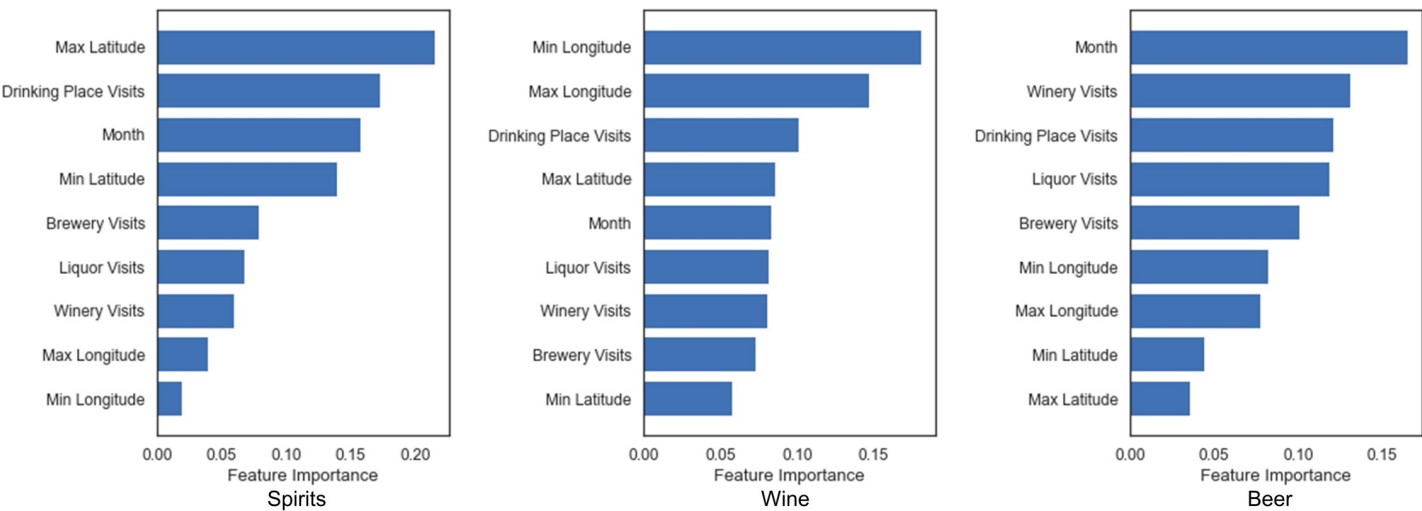

**Fig 14. Feature importance of the numeric input variables in the optimized RF models for the estimation of spirits, wine, and beer sales.**

three types of alcohol. Interestingly, visits to wineries seem to be an important feature for predicting beer sales. This could be due to the fact that certain states, such as OR, have strong sales in both wine and beer, but will need further investigation. It is also worth noting that the RF models were trained on data pre-COVID, and therefore the result in Fig 14 should be interpreted as the relative importance of the features during normal times. With more alcohol sales data, we could also examine the changed feature importance during COVID-19.

## The role of spatial and temporal features for alcohol sales estimation

In answering the three RQs, we have shown that alcohol sales, alcohol outlet visits, and their relations vary across different geographic areas. The feature importance analysis also emphasizes the role of spatial and temporal features for estimating alcohol sales. Here, we further examine the effect of three of these features, i.e., maximum latitude, minimum longitude, and month (which have the highest feature importance in their models), for estimating the sales of spirits, wine, and beer. Specifically, we performed sensitivity tests by computing the partial dependence between each of the three features and their corresponding target variable (i.e., the sales of spirits, wine, and beer). Partial dependence plots can reveal the marginal effect of a feature on the target variable, and the results of the three spatial and temporal features are shown in Fig 15. As can be seen, the sales of spirits generally increase with the increase of latitudes, while the sales of wine vary along longitude, with states close to the east and west coasts showing higher wine sales. This result is also consistent with the sales pattern of individual states revealed by our previous analysis in Section 3.1. In contrast, the sales of beer seem to be less affected by geography (as demonstrated by the relatively less importance of spatial features in Fig 14) but more affected by the month of a year: the sales of beer increase in the summer months and decrease during the winter.

## Implications to public health

The changes to alcohol purchasing behavior during the pandemic have implications for various societal concerns including public health policy, psychological health, addiction treatment, public disorder, and law enforcement. Alcohol sales policy varied considerably across states during the COVID-19 pandemic and the related closures with some states shutting down all

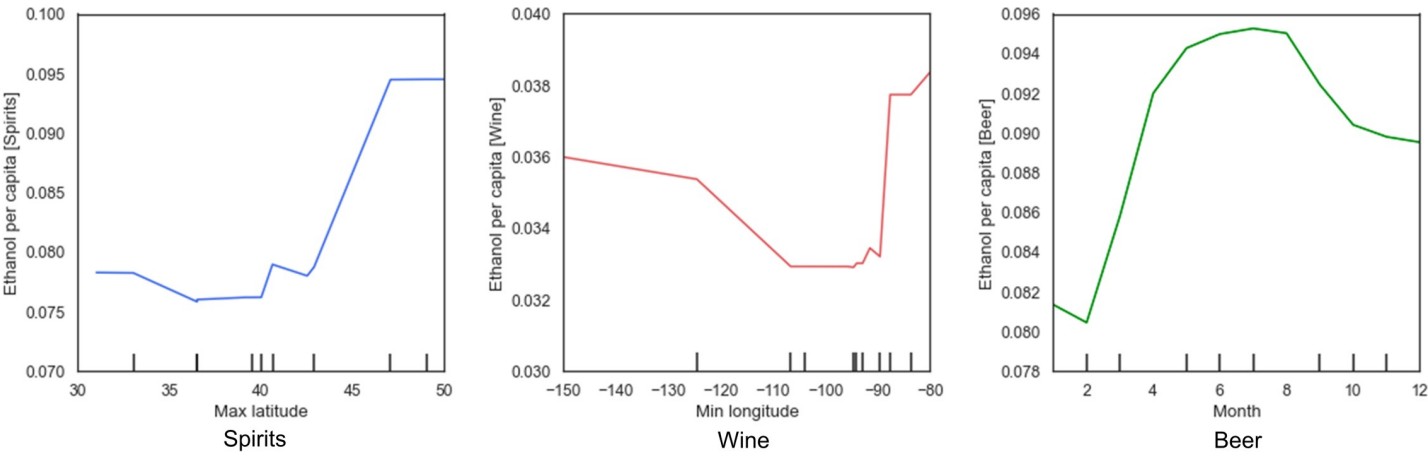

**Fig 15.** Partial dependence plots: (a) maximum latitude and spirits sales; (b) minimum longitude and wine sales; (c) month and beer sales.

types of alcohol sales and others only shutting down certain alcohol outlets such as bars. Our findings suggest that due to public policy changes during the pandemic (such as closing bars), beer sales went down but the volume of liquor and wine purchased increased. Related to our research, an online survey of buying behavior during the pandemic showed that approximately 38% of those responding to the survey indicated stockpiling of alcohol [42]. Although we do not know from our data whether increased alcohol purchasing led to increased consumption, emergent research has suggested that alcohol use increased during the pandemic among at least some segments of the population, especially those with comorbid psychological disorders such as anxiety and depression [10, 11].

Increased acute alcohol consumption is associated with a number of problems such as loss of work productivity, crime, and family violence [14, 43], and is generally associated with greater risk taking and impulsive behavior [44–46]. Additionally, the possible switch of individuals from lower-alcohol-content beverages such as beer to higher-alcohol-content beverages such as whiskey or vodka (even if drinking with the same frequency) can increase blood alcohol concentration more quickly and result in more severe acute pharmacological effects on behavior and cognitive functioning. Although some evidence proposed that hard liquor is more likely to promote negative social behavior [47], other research has suggested that intoxicated behavior can be better predicted using an individual's attributes rather than the type of alcohol [48, 49]. None of these issues were seemingly taken into consideration in terms of alcohol policy regarding public health edicts meant to combat the pandemic [6], and our findings suggest the need for a more comprehensive policy relating to alcohol availability, as to whether it should be considered an "essential" product, and whether access to alcohol of different varieties may have unintended deleterious effects on public health. While the primary purpose of the business shutdowns and stay-at-home orders was to reduce exposure to the virus, they can unintentionally result in greater alcohol consumption and may have the counterintuitive effect of increasing the exposure of some population groups to COVID-19 due to their risky behavior associated with more frequent or more acute intoxication. The changes we demonstrated regarding alcohol sales and visits to alcohol outlets suggest that public policies during times of pandemic may need to consider alcohol availability as a factor that influences public health above and beyond the reduction of exposure to COVID-19 at public drinking establishments [50].

### Limitations and future work

Recall that our analysis is based on the 16 U.S. states in which the NIAAA monthly alcohol sales data are available. Some states with high populations, such as California and New York, are not included in this dataset, which limits the conclusions we can draw. According to NIAAA, the monthly alcohol sales data were collected from various state sources that monitor alcohol sales primarily for taxation purposes [24], and such a data collection process can cost substantial financial and human resources. If more alcohol sales data were made available (including more frequently sampled data), our analysis could be expanded to other states and include more time samples, since the human mobility data from SafeGraph already covers the entire US and includes daily information.

At the same time, the limited NIAAA data also highlights an urgent need for the research community to develop more cost-effective approaches for collecting alcohol sales data covering large geographic areas and with fine spatial and temporal resolutions (e.g., daily alcohol sales data for the entire US at the county level). A complementary approach could utilize non-traditional data sources to estimate alcohol sales and supplement NIAAA's direct measurements. Importantly, the research that we presented herein highlights the value of human mobility data for broadening our understanding of alcohol sales. Future work could explore additional types of data that were also not created to specifically study alcohol, but which can be utilized to obtain insights about alcohol sales. One possible example would be to leverage transaction data from grocery stores if they become available. While our current analysis showed that grocery store visits cannot directly contribute to alcohol sales prediction, including transaction data could help us estimate the grocery store visits that involve alcohol purchase and further improve alcohol sales prediction.

## Conclusions

In this paper, we examined alcohol sales and alcohol outlet visits in 16 U.S. states before and during COVID-19. Motivated by anecdotal reports that alcohol sales surged after the pandemic began, we conducted empirical analyses based on the monthly alcohol sales data from NIAAA and alcohol outlet visits derived from the human mobility data provided by Safe-Graph. We focused on three research questions about alcohol sales, alcohol outlet visits, and their relation, and we leveraged various data analysis techniques and machine learning models to understand their changes during the pandemic and related geographic differences. Our findings showed that the sales of spirits and wine indeed surged since March 2020, but the sales of beer decreased. The per capita visits to liquor stores increased from April to June, but the visits to drinking places, breweries, and wineries dramatically decreased from March to June 2020. The relation between alcohol sales and alcohol outlet visits is complex, and we observed different behavior changes for different types of alcohol. Geographic differences were observed in alcohol sales, outlet visits, and their relation, suggesting people in different states changed their alcohol purchasing behavior differently in the face of the pandemic. While this study is not without limitations, it helps to reveal how COVID-19 and related public policies affected people's visits to alcohol outlets and their alcohol purchasing behavior.

## Supporting information

**S1 Fig. Wine sales in each of the 13 states.** The vertical dashed lines indicate the month of March in 2018, 2019, and 2020.
(TIF)

**S2 Fig. Beer sales in each of the 11 states.** The vertical dashed lines indicate the month of March in 2018, 2019, and 2020.
(TIF)

**S3 Fig. Change in per capita visits to drinking places in each state from March to June in 2020.** For each month, the percentage change is relative to the average value for the same month in 2018 and 2019.
(TIF)

**S4 Fig. Change in per capita visits to breweries in each state from March to June in 2020.** For each month, the percentage change is relative to the average value for the same month in 2018 and 2019.
(TIF)

**S5 Fig. Change in per capita visits to wineries in each state from March to June in 2020.** For each month, the percentage change is relative to the average value for the same month in 2018 and 2019.
(TIF)

**S6 Fig. Predicted and observed wine sales for individual states.** The vertical dashed lines indicate the month of March in 2018, 2019, and 2020.
(TIF)

**S7 Fig. Predicted and observed beer sales for individual states.** The vertical dashed lines indicate the month of March in 2018, 2019, and 2020.
(TIF)

## Author Contributions

**Conceptualization:** Yingjie Hu, Brian M. Quigley, Dane Taylor.

**Data curation:** Yingjie Hu.

**Formal analysis:** Yingjie Hu.

**Methodology:** Yingjie Hu, Brian M. Quigley, Dane Taylor.

**Validation:** Yingjie Hu, Brian M. Quigley, Dane Taylor.

**Visualization:** Yingjie Hu.

**Writing – original draft:** Yingjie Hu.

**Writing – review & editing:** Yingjie Hu, Brian M. Quigley, Dane Taylor.

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
