## [Decision Letter · Decision Letter 0]

6 May 2021

PONE-D-21-09766

Human mobility data and machine learning reveal geographic differences in U.S. alcohol sales and alcohol outlet visits during COVID-19

PLOS ONE

Dear Dr. Hu,

Thank you for submitting your manuscript to PLOS ONE. After careful consideration of reviewers, we thought that the paper has merit but requires a major revision to meet PLOS ONE’s publication criteria as it currently stands. Therefore, we invite you to submit a revised version of the manuscript that addresses the points raised during the review process.

We look forward to receiving your revised manuscript.

Kind regards,

Song Gao, Ph.D.

Academic Editor

PLOS ONE

Journal Requirements:

2. We note that Figures 4, 6, 8, 11, S3, S4 and S5 in your submission contain map images which may be copyrighted.

a. You may seek permission from the original copyright holder of Figures 4, 6, 8, 11, S3, S4 and S5 to publish the content specifically under the CC BY 4.0 license. 

Reviewers' comments:

Reviewer's Responses to Questions

**Comments to the Author**

1. Is the manuscript technically sound, and do the data support the conclusions?

Reviewer #1: Yes

Reviewer #2: No

2. Has the statistical analysis been performed appropriately and rigorously? 

Reviewer #1: Yes

Reviewer #2: No

3. Have the authors made all data underlying the findings in their manuscript fully available?

Reviewer #1: Yes

Reviewer #2: Yes

4. Is the manuscript presented in an intelligible fashion and written in standard English?

Reviewer #1: Yes

Reviewer #2: Yes

5. Review Comments to the Author

Reviewer #1: Review of Human mobility data and machine learning reveal geographic differences in U.S. alcohol sales and alcohol outlet visits during COVID-19

This article demonstrates the relationships between alcohol sales and outlet visits during the COVID-19. Results show that the alcohol sales are different regards types of alcohols and geographic regions. People reduce their visits to alcohol outlets except for liquor stores. Machine learning models are trained to examine the relationships between alcohol sales and outlet visits. It is overall an interesting paper and is valuable to inform the policymaking. Before its publication, several issues should be addressed as follows.

1. The figures are overall very nice. While some months should be highlighted not only in March 2020 but also in March 2019, 2018. I would suggest remove the grid lines of the background but highlight those months in Figure 2, 3, 5, 10, 13.

2. The authors mentioned that they performed the ten-fold validation when choosing the best models. However, for time series forecasting, it might be better to perform the “walk-forward approach” for validation while not the traditional cross-validation.

3. The authors use machine learning to answer the RQ3 about the relationship between alcohol sales and outlet visits. The authors should add more content to clarify why it is necessary to use machine learning models (which are usually used for non-linear relationships) while not some linear regression or correlation analysis to reflect such relationships.

4. Line 480: it makes sense that RF performs better than the DNN model as the latter one usually achieves better accuracies in some more complex tasks such as computer vision and natural language processing. Hence, it is not a “surprise”.

Reviewer #2: The authors examined alcohol sales and alcohol outlet visits in 16 U.S. states using surveillance reports from the National Institute on Alcohol Abuse and POI visits from various alcohol outlets released by SafeGragh, a commercial company. They further examined geographic differences in changes to alcohol sales and outlet visits during the COVID-19 stay-at-home period. Their results suggest increases in sales of spirits and wine since March 2020, while decreases in the sales of beer. This manuscript is written in great English and is easy to follow. However, the reviewer observed several major limitations that impact the scientific value of this research.

1. The authors only investigated 16 U.S. states in U.S. (out of 50) due to data limitations. Several heavily populated states, such as NY and CA, are not even included. Such a limitation greatly reduces the scientific value of this research, given the small and presumably biased spatial coverage of their data. The author mentioned “geographic difference” many times (also appeared in their title). However, besides presenting the spatial distribution in maps, no spatial algorithm is implemented, nor connection to geographic laws is made. The geographic knowledge is presented by the effort of “mapping”, and “mapping” only.

2. The authors selected four types of POI as alcohol outlets in their study, i.e., Liquor Store, Drinking Place, Brewery, and Winery. However, as the authors mentioned, people can still purchase alcohol from other types of POIs, such as grocery stores, which the reviewer believes is actually the major source for alcohol purchase. Due to the pandemic, it is reasonable to assume that people might reduce the diversity of their POI visits (such as reducing visits to Drinking Places, Brewery, etc.) and only keep essential ones. Grocery store visiting is essential, and it is likely people will direct their alcohol purchases in grocery stores. Unfortunately, such a major (influential) alcohol outlet is not covered.

3. The authors tested three algorithms multiple regression, Random Forest, and DNN. For Multiple Regression, no collinearity test is presented to reveal if these input parameters satisfy the underlying independent assumption. For the Random Forest model, no hyperparameter tuning is implemented to search for the best parameter configuration (the author just used 100 trees as default and did not explore other optimization options). The authors did not provide an evaluation of variable importance nor a sensitivity test to reveal how different inputs would influence their prediction.

6. PLOS authors have the option to publish the peer review history of their article (what does this mean?). If published, this will include your full peer review and any attached files.

Reviewer #1: No

Reviewer #2: No

---

## [Author Response · Author response to Decision Letter 0]

14 Jun 2021

Reviewer #1:

This article demonstrates the relationships between alcohol sales and outlet visits during the COVID-19. Results show that the alcohol sales are different regards types of alcohols and geographic regions. People reduce their visits to alcohol outlets except for liquor stores. Machine learning models are trained to examine the relationships between alcohol sales and outlet visits. It is overall an interesting paper and is valuable to inform the policymaking. Before its publication, several issues should be addressed as follows.

Our response: We sincerely thank the reviewer for reading our manuscript and giving us constructive comments. Please find our item-by-item responses below.

1. The figures are overall very nice. While some months should be highlighted not only in March 2020 but also in March 2019, 2018. I would suggest remove the grid lines of the background but highlight those months in Figure 2, 3, 5, 10, 13.

Our response: We thank the reviewer for this suggestion. In this revision, we highlighted the months of March 2018, 2019, and 2020, and removed the vertical grid lines of the background. We applied this change to all figures with time series data, which are Figure 2, 3, 5, 7, 9, 10, 13, S1, S2, S6, and S7.

2. The authors mentioned that they performed the ten-fold validation when choosing the best models. However, for time series forecasting, it might be better to perform the “walk-forward approach” for validation while not the traditional cross-validation.

Our response: Following the reviewer’s suggestion, we performed walk-forward validation by dividing the training and validation data based on months and letting the model predict three months ahead. The performance of all three models slightly improved in the walk-forward validation while random forest still showed the best performance. The experiment result is summarized in Figure 12. We also added a relevant description on walk-forward validation in Lines 251-255: “Considering that the models will be applied to predicting future alcohol sales, we also used walk-forward validation which is specifically suitable for time series data (40). Starting from a model trained using the data in the whole year of 2018, we validated the model using the data in the next three months and walked the model forward month by month with the trained model always predicting alcohol sales in the next three months.” We added a discussion on the experiment results in Lines 501-506: “The performances of the three models in estimating alcohol sales are shown in Fig 12, with the first row showing the result of ten-fold cross-validation and the second row showing the result of walk-forward validation. RMSE was calculated based on the predicted alcohol sales and the recorded sales in the NIAAA data. The two sets of validation experiments showed similar results: all three models provided reasonable accuracies, while the RF model had the best performance among the three, as demonstrated by its lowest RMSE.”

3. The authors use machine learning to answer the RQ3 about the relationship between alcohol sales and outlet visits. The authors should add more content to clarify why it is necessary to use machine learning models (which are usually used for non-linear relationships) while not some linear regression or correlation analysis to reflect such relationships.

Our response: We thank the reviewer for this comment. We have now added a clarification in Lines 184-187: “We used random forest and deep neural networks, rather than linear regression alone, because these two machine learning models can effectively capture the complex and nonlinear relationships between the input features and the target variable (e.g., nonstationary trends and periodicity).”

4. Line 480: it makes sense that RF performs better than the DNN model as the latter one usually achieves better accuracies in some more complex tasks such as computer vision and natural language processing. Hence, it is not a “surprise”.

Our response: We thank the reviewer for the suggestion and agree that given past findings this is not necessarily a surprise in our data. We have revised the sentences accordingly and removed the part related to “surprise” (in Lines 505-508): “...while the RF model had the best performance among the three, as demonstrated by its lowest RMSE. The RF model also performed better than the more complicated DNN model. A possible explanation is that this prediction task is based on a small training dataset ...”

 

Reviewer #2:

The authors examined alcohol sales and alcohol outlet visits in 16 U.S. states using surveillance reports from the National Institute on Alcohol Abuse and POI visits from various alcohol outlets released by SafeGragh, a commercial company. They further examined geographic differences in changes to alcohol sales and outlet visits during the COVID-19 stay-at-home period. Their results suggest increases in sales of spirits and wine since March 2020, while decreases in the sales of beer. This manuscript is written in great English and is easy to follow. However, the reviewer observed several major limitations that impact the scientific value of this research.

Our response: We sincerely thank the reviewer for reading our manuscript and giving us constructive comments. Please find our item-by-item responses below.

1a. The authors only investigated 16 U.S. states in U.S. (out of 50) due to data limitations. Several heavily populated states, such as NY and CA, are not even included. Such a limitation greatly reduces the scientific value of this research, given the small and presumably biased spatial coverage of their data.

Our response: We thank the reviewer’s comment. We agree with the reviewer that the limited alcohol sales data available from NIAAA is a limitation of our work. At the same time, the lack of alcohol sales data also highlights an urgent need for the research community to develop more cost-effective approaches for data collection. In this revision, we enhanced our discussion on both points, as follows: 

First, we further acknowledged this limitation following the reviewer’s comment (in Lines 674-682): “… our analysis is based on the 16 U.S. states in which the NIAAA monthly alcohol sales data are available. Some states with high populations, such as California and New York, are not included in this dataset, which limits the conclusions we can draw. According to NIAAA, the monthly alcohol sales data were collected from various state sources that monitor alcohol sales primarily for taxation purposes (24), and such a data collection process can cost substantial financial and human resources. If more alcohol sales data were made available (including more frequently sampled data), our analysis could be expanded to other states and include more time samples, since the human mobility data from SafeGraph already covers the entire US and includes daily information.” 

Second, we discussed the need for the research community to develop more cost-effective approaches for data collection (in Lines 683-694): “At the same time, the limited NIAAA data also highlights an urgent need for the research community to develop more cost-effective approaches for collecting alcohol sales data covering large geographic areas and with fine spatial and temporal resolutions (e.g., daily alcohol sales data for the entire US at the county level). A complementary approach could utilize non-traditional data sources to estimate alcohol sales and supplement NIAAA’s direct measurements. Importantly, the research that we presented herein highlights the value of human mobility data for broadening our understanding of alcohol sales. Future work could explore additional types of data that were also not created to specifically study alcohol, but which can be utilized to obtain insights about alcohol sales. One possible example would be to leverage transaction data from grocery stores if they become available. While our current analysis showed that grocery store visits cannot directly contribute to alcohol sales prediction, including transaction data could help us estimate the grocery store visits that involve alcohol purchase and further improve alcohol sales prediction.”

1b. The author mentioned “geographic difference” many times (also appeared in their title). However, besides presenting the spatial distribution in maps, no spatial algorithm is implemented, nor connection to geographic laws is made. The geographic knowledge is presented by the effort of “mapping”, and “mapping” only.

Our response: The reviewer has rightfully pointed out an important part of this work which needed greater emphasis. In this revision, we further analyzed the role of spatial and temporal features (i.e., latitude, longitude, and month) in affecting the sales of different types of alcohol. We found that the sales of spirits are largely affected by the latitude of a state, i.e., states in higher latitudes typically have higher spirits sales. Meanwhile, the sales of wine in the US are largely affected by the longitude of state, with states close to the east and west coasts having higher wine sales than states in the middle. In contrast, the sales of beer do not seem to be clearly affected by geography but by the month of a year, i.e., summer months have higher sales than winter months. This revision can be seen in the newly added Section “The role of spatial and temporal features for alcohol sales estimation” in Lines 620-638: “In answering the three RQs, we have shown that alcohol sales, alcohol outlet visits, and their relations vary across different geographic areas. The feature importance analysis also emphasizes the role of spatial and temporal features for estimating alcohol sales. Here, we further examine the effect of three of these features, i.e., maximum latitude, minimum longitude, and month (which have the highest feature importance in their models), for estimating the sales of spirits, wine, and beer. ...”

2. The authors selected four types of POI as alcohol outlets in their study, i.e., Liquor Store, Drinking Place, Brewery, and Winery. However, as the authors mentioned, people can still purchase alcohol from other types of POIs, such as grocery stores, which the reviewer believes is actually the major source for alcohol purchase. Due to the pandemic, it is reasonable to assume that people might reduce the diversity of their POI visits (such as reducing visits to Drinking Places, Brewery, etc.) and only keep essential ones. Grocery store visiting is essential, and it is likely people will direct their alcohol purchases in grocery stores. Unfortunately, such a major (influential) alcohol outlet is not covered.

Our response: We thank the reviewer for this comment. The fact that individuals can also purchase alcohol from grocery stores was a consideration when we initially approached the data, although we chose to leave it out of the first round of analyses. We initially didn’t include grocery store visits into the input features because we felt that the link between a grocery store visit and alcohol purchase is weaker than the POIs we used, all of which exist for the main purpose of selling alcohol. Nevertheless, in this revision, we did additional experiments to empirically examine the effect of including grocery store visits for alcohol sales prediction. We compared the performance of the RF models that included and not included grocery store visits as one of the input features. The result showed that there is almost no change in the performance of the model when grocery store visits are included. This result is now added into the paper as Table 3, and we also added a discussion on this experiment in Lines 521-532: “In our selection of alcohol outlet types, we excluded grocery stores because the link between a grocery store visit and alcohol purchase is unclear. Nevertheless, people can purchase alcohol from grocery stores that remained open during the shutdown period. Thus, we further examined the effect of including grocery store visits as an additional predictor for alcohol sales. We used the RF model for this examination which showed the best performance among the three models. Table 3 summarizes the RMSE values of the RF models which included or not included grocery store visits as one of their input features. The result shows that there is almost no change in the performance of the model when grocery store visits are included. In fact, the RMSE values became slightly worse for wine and beer sales estimations when grocery store visits were included. The experiment result suggests that grocery store visits may be too noisy to effectively contribute to alcohol sales estimation. While people can indeed purchase alcohol from grocery stores, there also exist many grocery store visits which do not involve alcohol purchase.”

3. The authors tested three algorithms multiple regression, Random Forest, and DNN. For Multiple Regression, no collinearity test is presented to reveal if these input parameters satisfy the underlying independent assumption. For the Random Forest model, no hyperparameter tuning is implemented to search for the best parameter configuration (the author just used 100 trees as default and did not explore other optimization options). The authors did not provide an evaluation of variable importance nor a sensitivity test to reveal how different inputs would influence their prediction.

Our response: We thank the reviewer for these constructive comments. We made the following revisions following your comments. First, for MLR, we performed multicollinearity tests by calculating the variance inflation factor (VIF) for the numeric independent variables and gradually removing the variables with the highest VIF (until all VIF values are smaller than the cut-off value 5). The results of the multicollinearity tests are summarized in Table 2, and related discussion is added in Lines 489-498: “For MLR, we first performed multicollinearity tests by calculating the variance inflation factor (VIF) for the numeric independent variables. Table 2 shows the VIF values for three MLR models in which we gradually removed the variable with the highest VIF value. The test identified collinearity between minimum and maximum latitudes and longitudes, which can be expected. By removing maximum latitude and then maximum longitude, we obtained Reduced Model 2 whose VIF values are all smaller than the typically cut-off value 5. However, the RMSE values of the three models are the same...”

Second, we performed hyperparameter tuning to identify the optimized parameter configuration for the RF model. This revision can be seen in Lines 535-543: “Next, we tune the hyperparameters of the RF model to identify the optimized parameter configuration. We performed hyperparameter tuning for spirits, wine, and beer respectively. Learning from the literature (31), we focused on tuning two major hyperparameters, namely n_tree, which controls the number of decision trees, and m_try, which controls the number of features to consider at each split in a tree. We performed grid search to identify the best hyperparameters, and the search space for n_tree is set to [10, 200] with an interval of 10, and the search space for m_tryis {S,√S,log_2 S,S/2,S/3}where S is the number of input features. We limited the search space for the number of trees to 200 because the dataset is relatively small. The identified optimized hyperparameters were then used in the final RF models for alcohol sales estimation.” In addition, we updated Figs 13, S6, and S7 based on the estimates of the optimized RF models. 

Third, we analyzed the feature importance output by the optimized RF models. The feature importances are summarized in a new figure, Fig 14, and we added a related discussion in Lines 599-616: “The use of the random forest model allows us to examine the relative importance of the input features for alcohol sales estimation. ... As can be seen, spatial and temporal features, namely maximum latitude, minimum longitude, and month, play highly important roles in helping the model estimate alcohol sales ... While the relative importance of other features varies across alcohol types, visits to drinking places are an important feature for predicting the sales of spirits, wine, and beer. In addition, visits to liquor stores also have moderate importance across all three types of alcohol...” We also performed sensitivity tests by computing the partial dependence between latitude, longitude, and month and different types of alcohol sales. The partial dependence result is summarized in a new figure, Fig 15, and we also added a related discussion in Lines 623-635: “Here, we further examine the effect of three of these features, i.e., maximum latitude, minimum longitude, and month (which have the highest feature importance in their models), for estimating the sales of spirits, wine, and beer. Specifically, we performed sensitivity tests by computing the partial dependence between each of the three features and their corresponding target variable (i.e., the sales of spirits, wine, and beer). Partial dependence plots can reveal the marginal effect of a feature on the target variable, and the results of the three spatial and temporal features are shown in Fig 15. As can be seen, the sales of spirits generally increase with the increase of latitudes...”

---

## [Decision Letter · Decision Letter 1]

27 Aug 2021

PONE-D-21-09766R1

Human mobility data and machine learning reveal geographic differences in U.S. alcohol sales and alcohol outlet visits during COVID-19

PLOS ONE

Dear Dr. Hu,

Thank you for submitting your manuscript to PLOS ONE. After careful consideration, we feel that it has merit but does not fully meet PLOS ONE’s publication criteria as it currently stands. Therefore, we invite you to submit a revised version of the manuscript that addresses the points raised during the review process.

A rebuttal letter that responds to each point raised by the academic editor and reviewer(s). You should upload this letter as a separate file labeled 'Response to Reviewers'.A marked-up copy of your manuscript that highlights changes made to the original version. You should upload this as a separate file labeled 'Revised Manuscript with Track Changes'.An unmarked version of your revised paper without tracked changes. You should upload this as a separate file labeled 'Manuscript'

We look forward to receiving your revised manuscript.

Kind regards,

Johnson Chun-Sing Cheung, D.S.W.

Academic Editor

PLOS ONE

Journal Requirements:

Reviewers' comments:

Reviewer's Responses to Questions

**Comments to the Author**

1. If the authors have adequately addressed your comments raised in a previous round of review and you feel that this manuscript is now acceptable for publication, you may indicate that here to bypass the “Comments to the Author” section, enter your conflict of interest statement in the “Confidential to Editor” section, and submit your "Accept" recommendation.

Reviewer #1: All comments have been addressed

Reviewer #2: (No Response)

Reviewer #3: All comments have been addressed

2. Is the manuscript technically sound, and do the data support the conclusions?

Reviewer #1: Yes

Reviewer #2: Yes

Reviewer #3: Yes

3. Has the statistical analysis been performed appropriately and rigorously? 

Reviewer #1: Yes

Reviewer #2: Yes

Reviewer #3: Yes

4. Have the authors made all data underlying the findings in their manuscript fully available?

Reviewer #1: Yes

Reviewer #2: Yes

Reviewer #3: Yes

5. Is the manuscript presented in an intelligible fashion and written in standard English?

Reviewer #1: Yes

Reviewer #2: Yes

Reviewer #3: Yes

7. PLOS authors have the option to publish the peer review history of their article (what does this mean?). If published, this will include your full peer review and any attached files.

Reviewer #1: No

Reviewer #2: No

Reviewer #3: No

6. Review Comments to the Author

Reviewer #2: The authors enhance the discussion on their data issues. However, the reviewer believes that their dataset that only covers 16 U.S. states out of 50 is a huge mismatch to their title, which is "revealing geographic differences in U.S. alcohol sales and alcohol outlet visits". How can U.S. 16 states (without heavily populated states like NY and CA) represent the entire U.S? If this limitation can not be addressed, at least, the title of this study needs to be modified to correctly reflect the scope of this study. "in (16) selected U.S. States" could be an option. The reviewer is satisfied with other responses.

Reviewer #3: Thank you for addressing appropriately to most of the comments given by reviewers. Before I can recommend this paper for publication, there is still one thing I would like to point out regarding the result of this study. Authors attempted to apply a number of cutting-edge methodology including human mobility data and machine learning in this study, so as to study the association between alcohol sales data and visiting behavior of people to different POIs including alcohol outlets. The result section of this important area remains unclear. In a nutshell, do authors argue that “if more people visit those POIs, it resulted in the increasing of alcohol sale”? (if so, it sounds quite commonsensical) If not, what the authors were trying to say after investigating their associations? In page 19, authors highlighted that “While the primary purpose of the business shutdowns and stay-at-home orders was to reduce exposure to the virus, they can unintentionally result in greater alcohol consumption and may have the counterintuitive effect of increasing the exposure of some population groups to COVID-19 due to their risky behavior associated with more frequent or more acute intoxication. The changes we demonstrated regarding alcohol sales and visits to alcohol outlets suggest that public policies during times of pandemic may need to consider alcohol availability as a factor that influences public health above and beyond the reduction of exposure to COVID-19 at public drinking establishments.” How do the findings (particularly related to RQ3) point to the aforementioned conclusion? All in all, what are the important implications that authors would try to contribute after having incorporating those cutting-edge methodology, namely human mobility data and machine learning?

---

## [Author Response · Author response to Decision Letter 1]

10 Sep 2021

Responses to review comments:

Reviewer #2: 

The authors enhance the discussion on their data issues. However, the reviewer believes that their dataset that only covers 16 U.S. states out of 50 is a huge mismatch to their title, which is "revealing geographic differences in U.S. alcohol sales and alcohol outlet visits". How can U.S. 16 states (without heavily populated states like NY and CA) represent the entire U.S? If this limitation can not be addressed, at least, the title of this study needs to be modified to correctly reflect the scope of this study. "in (16) selected U.S. States" could be an option. The reviewer is satisfied with other responses.

Our response: We thank the reviewer’s further comment. We carefully considered the suggestion of the reviewer on revising the title to “in (16) selected U.S. States”, but felt that the word “selected” might imply that we deliberately chose a subset of the available data (while we studied all the available data). We eventually revised the title to “Human mobility data and machine learning reveal geographic differences in alcohol sales and alcohol outlet visits across U.S. states during COVID-19”. In addition, our abstract also explicitly states that we study a subset of states. 

Reviewer #3:

Thank you for addressing appropriately to most of the comments given by reviewers. Before I can recommend this paper for publication, there is still one thing I would like to point out regarding the result of this study. 

Our response: Many thanks to the reviewer for reading our paper and giving us constructive comments. Please see our item-by-item responses below.

Authors attempted to apply a number of cutting-edge methodology including human mobility data and machine learning in this study, so as to study the association between alcohol sales data and visiting behavior of people to different POIs including alcohol outlets. The result section of this important area remains unclear. In a nutshell, do authors argue that “if more people visit those POIs, it resulted in the increasing of alcohol sale”? (if so, it sounds quite commonsensical) If not, what the authors were trying to say after investigating their associations? 

Our response: We thank the reviewer’s comment. Indeed, in a normal situation, more visits to alcohol outlets will likely lead to higher alcohol sales. However, under the special context of COVID-19, higher alcohol sales are not necessarily linked to more alcohol outlet visits, since people may purchase alcohol online in order to reduce their exposure to the virus, or people may purchase an increased amount of alcohol in a single visit. Our study, particularly RQ3, examines how the relation between alcohol sales and alcohol outlet visits might have changed before and during the early months of COVID-19. 

In this revision, we clarified this point in Lines 570-584: “While alcohol sales and alcohol outlet visits are naturally related, our study focused on how their relation might have changed under the special context of the COVID-19 pandemic. We trained three types of machine learning models using the pre-COVID data and then compared model estimates with the recorded alcohol sales in March, April, May, and June 2020. The results based on the best RF models showed large deviations between model estimates and the recorded alcohol sales in some states and particularly in the month of March 2020. These large deviations suggest that the relation between alcohol sales and alcohol outlet visits might have changed in ways that are no longer captured by the trained model (e.g., due to people purchasing alcohol online or purchasing a large amount of alcohol in one visit). Such large deviations were observed in spirits and wine sales, but not in beer sales whose model estimates were still fairly close to the recorded data. In addition, geographic differences were again observed in this relation change, with stronger changes observed in some states (e.g., spirits sales in AK, FL, MO, and ND) but not in some other states (e.g., spirits sales in AR, IL, KY, and TX,). More research is necessary to further understand how exactly people might have changed their alcohol purchasing behavior.”

In page 19, authors highlighted that “While the primary purpose of the business shutdowns and stay-at-home orders was to reduce exposure to the virus, they can unintentionally result in greater alcohol consumption and may have the counterintuitive effect of increasing the exposure of some population groups to COVID-19 due to their risky behavior associated with more frequent or more acute intoxication. The changes we demonstrated regarding alcohol sales and visits to alcohol outlets suggest that public policies during times of pandemic may need to consider alcohol availability as a factor that influences public health above and beyond the reduction of exposure to COVID-19 at public drinking establishments.” How do the findings (particularly related to RQ3) point to the aforementioned conclusion? 

Our response: We thank the reviewer for noticing this issue. In this version, we revised the text to enhance the connection between this conclusion and our experiment results (see Lines 649-661): “While the primary purpose of the business shutdowns and stay-at-home orders was to reduce exposure to the virus, they can also result in unintentional consequences, such as increased spirits and wine sales (as shown in Figs. 2 and 5) and increased visits to liquor stores (as shown in Fig. 9). Our geographical analysis also shows that the increases in spirits and wine sales were as large as 20-40% in multiple states (as shown in Figs. 4 and 6). These increased alcohol sales and liquor store visits can in turn lead to greater alcohol consumption and may even increase the exposure of some population groups to COVID-19 due to their increased visits to liquor stores. Additionally, if these increased liquor sales are indicative of increased drinking, individuals engaging in heavy drinking may be less likely to engage in safe behavioral strategies, such as social distancing and mask wearing, that are intended to protect them from the exposure to the virus (1,44). Our study suggests that public policies during a pandemic like COVID-19 may need to consider alcohol availability as a risk factor in addition to others that can affect public health.”

All in all, what are the important implications that authors would try to contribute after having incorporating those cutting-edge methodology, namely human mobility data and machine learning?

Our response: We thank the reviewer’s comment. In this version, we revised and enhanced our description of the contributions of this work in Lines 691-704: “Our contributions focused on three research questions regarding alcohol sales, alcohol outlet visits, and their relation respectively. First, our analysis showed some consistency with anecdotal reports in that the sales of spirits and wine indeed significantly increased since March 2020 compared to the same months in previous years; however, the increase was not homogeneous across different alcohol types, and in fact, the sales of beer largely decreased. Second, we found that people’s visits to liquor stores largely increased in the early months of the pandemic, but visits to other types of alcohol outlets (i.e., drinking places, breweries, and wineries) dramatically decreased. Third, we found that the relation between alcohol sales and alcohol outlet visits might have changed after the pandemic began in ways that can no longer be captured by machine learning models that were trained on alcohol outlet visits and other general information, due to possible reasons such as online alcohol purchase and stockpiling. In addition, our analysis showed geographic differences in alcohol sales, outlet visits, and their relation, which revealed the different responses to COVID-19 in terms of people’s alcohol purchasing behavior.”

---

## [Decision Letter · Decision Letter 2]

17 Nov 2021

Human mobility data and machine learning reveal geographic differences in alcohol sales and alcohol outlet visits across U.S. states during COVID-19

PONE-D-21-09766R2

Dear Dr. Hu,

We’re pleased to inform you that your manuscript has been judged scientifically suitable for publication and will be formally accepted for publication once it meets all outstanding technical requirements.

Editor's note:

*For your information, I have served as reviewer #3 of this paper so as to expedite the review process. Thank you for your attention.*

Kind regards,

Johnson Chun-Sing Cheung, D.S.W.

Academic Editor

PLOS ONE

---

## [Editor Report · Acceptance letter]

17 Nov 2021

PONE-D-21-09766R2 

Human mobility data and machine learning reveal geographic differences in alcohol sales and alcohol outlet visits across U.S. states during COVID-19 

Dear Dr. Hu:

I'm pleased to inform you that your manuscript has been deemed suitable for publication in PLOS ONE. Congratulations! Your manuscript is now with our production department. 

Kind regards, 

on behalf of

Dr. Johnson Chun-Sing Cheung 

Academic Editor

PLOS ONE